# Inductive Global and Local Manifold Approximation and Projection

**Jungeum Kim**                                                    *Jungeum.kim@chicagobooth.edu*
*Booth School of Business*
*University of Chicago*

**Xiao Wang**                                                      *wangxiao@purdue.edu*
*Statistics Department*
*Purdue University*

**Reviewed on OpenReview:** *https://openreview.net/forum?id=p9pxeNupQ5&noteId=VEi3S62pV4*

## Abstract

Nonlinear dimensional reduction with the manifold assumption, often called manifold learning, has proven its usefulness in a wide range of high-dimensional data analysis. The significant impact of t-SNE and UMAP has catalyzed intense research interest, seeking further innovations toward visualizing not only the local but also the global structure information of the data. Moreover, there have been consistent efforts toward generalizable dimensional reduction that handles unseen data. In this paper, we first propose GLoMAP, a novel manifold learning method for dimensional reduction and high-dimensional data visualization. GLoMAP preserves locally and globally meaningful distance estimates and displays a progression from global to local formation during the course of optimization. Furthermore, we extend GLoMAP to its inductive version, iGLoMAP, which utilizes a deep neural network to map data to its lower-dimensional representation. This allows iGLoMAP to provide lower-dimensional embeddings for unseen points without needing to re-train the algorithm. iGLoMAP is also well-suited for mini-batch learning, enabling large-scale, accelerated gradient calculations. We have successfully applied both GLoMAP and iGLoMAP to the simulated and real-data settings, with competitive experiments against the state-of-the-art methods.

*Keywords*: Data visualization, deep neural networks, nonlinear dimensional reduction, inductive algorithm

## 1 Introduction

Data visualization, which belongs to exploratory data analysis, has been promoted by John Tukey since the 1970s as a critical component in scientific research (Tukey, 1977). However, visualizing high-dimensional data is a challenging task. The lack of visibility in the high-dimensional space gives less intuition about what assumptions would be necessary for compactly presenting the data on the reduced dimension. A common assumption that can be made without specific knowledge of the high-dimensional data is the manifold assumption. It assumes that the data are distributed on a low-dimensional manifold within a high-dimensional space. Nonlinear dimensional reduction (DR) with the manifold assumption, often called *manifold learning*, has proven its usefulness in a wide range of high-dimensional data analysis (Meilă & Zhang, 2023). Various research efforts have been made to develop DR tools for data visualization, and several leading algorithms include MDS (Cox & Cox, 2008), Isomap (Tenenbaum et al., 2000), t-SNE (Van der Maaten & Hinton, 2008), UMAP (McInnes et al., 2018), PHATE (Moon et al., 2017), PacMAP (Wang et al., 2021), and many others. It is not uncommon to perform further statistical analysis on the reduced dimensional data, including visualized data, through methods such as regression (Cheng & Wu, 2013), functional data analysis (Dai & Müller, 2018), classification (Belkina et al., 2019), and generative models (Qiu & Wang, 2021).

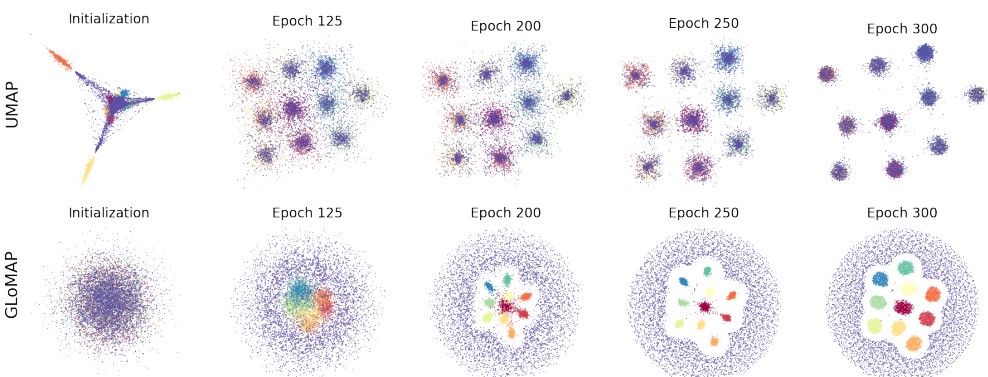

Figure 1: The visualization of the spheres dataset (Moor et al., 2020) by UMAP and GLoMAP (both transductive). GLoMAP shows a progression of the representation from global to local during the optimization. All ten inner clusters are identified as well as the larger cluster on the outer shell (purple).

The purpose of this work is to address a key challenge in manifold learning: Can a single algorithm capture both the global structures and local details of high-dimensional data? To answer this question, we develop a new nonlinear manifold learning method called GLoMAP, which is short for Global and Local Manifold Approximation and Projection. An essential component of GLoMAP is its locally adaptive global distance construction, where locality is based on $K$-nearest neighbors, and global distance refers to the distance between any general pair of data points. GLoMAP *locally* approximates the data manifold in a data-adaptive way with many small low-dimensional Euclidean patches similar to UMAP (McInnes et al., 2018). As such an approximation can result in multiple different local distances of the same pairs, we first reconcile the local incompatibilities by taking the maximum of the multiple (finite) local distances. Then, we take a shortest path search over those local maximums to obtain a coherent *global* metric. Our heuristic gives a single topological space that reflects the global geometry of the data manifold without incompatible distances.

Our work presents two significant algorithmic advancements. First, to address the crowding problem in visualization highlighted by Van der Maaten & Hinton (2008), we adopt a UMAP-like loss function. Optimizing this loss function with a global distance matrix (rather than a sparse matrix of $K$-nearest neighbors) presents a challenge. To handle the challenge, we develop a novel sampling scheme that leverages an unbiased estimator of the loss (Proposition 1) within a stochastic gradient descent framework. Second, our annealing-like process for scaling—similar to cooling in temperature-is a novel and interesting development. It enhances the optimization process by enabling a gradual progression from global shape to local detail. This progressive global-to-local refinement is unique in the literature and addresses a longstanding dilemma: global methods often lack local detail, while local methods rely on initialization for global coherence. Specifically, by tempering with a global distance rescaler $\tau$, the algorithm first identifies global structures (with a larger $\tau$) and subsequently refines local details (with a smaller $\tau$). This effect, illustrated in Figure 1, reveals a clear progression from global structure identification to fine-grained local details, where all ten inner clusters are distinguished, as well as the larger cluster on the outer shell (purple). We emphasize that this annealing approach does not require carefully crafted initialization—random noise suffices. We find that combining annealing with global distance scaling was particularly effective, as similar meaningful structures did not emerge with rescaled local distances alone.

To achieve an inductive dimensional reduction mapping for the generalization of visualization to an unseen novel data point, we extend GLoMAP to its inductive version named iGLoMAP (inductive GLoMAP). The low dimensional embedding, denoted by $z$, for a high dimensional data point $x$ is obtained by $z = Q_\theta(x)$, where $Q_\theta$ is modeled by a deep neural network (DNN) with parameters $\theta$. When the map $Q_\theta$ is updated by using a gradient from a pair of points, the low dimensional representation of the other pairs is also affected. We found that it is very difficult to optimize the GLoMAP loss function with a typical deep learning method where the visualization representations are by a neural network's output. Therefore, as one of our key contributions, we develop a particle-based algorithm which mimics the stable optimization process

of transductive learning. By this, we can formulate an effective stochastic gradient descent algorithm for learning the inductive map.

Our proposed global distance itself, constructed through a new local distance, also can be seen as an advance in the array of global preservation algorithms. MDS keeps the metric information between all the pairs of data points, and thereby aims to preserve the global geometry of the dataset. Recognizing that Euclidean distances between non-neighboring points may not always be informative in high-dimensional settings, Isomap applies MDS instead to geodesic distance estimates computed via the shortest path search. Efforts to refine this geodesic distance estimator include improving the Euclidean distances among nearby points to better reflect the local manifold, for instance, through a conformal embedding (Silva & Tenenbaum, 2002), a spherelets argument (Li & Dunson, 2019), or the tangential Delaunay complex (Arias-Castro & Chau, 2020). On the refined local distances, these approaches apply the shortest path search to estimate global distances. As our local distance estimator is in a closed form, the distance computation is a scalable alternative to Li & Dunson (2019) and Arias-Castro & Chau (2020). Similar to our approach, the local distance in Silva & Tenenbaum (2002) is computed by multiplying a closed-form local rescaling constant to the Euclidean distances. As compared in Section S9 in Appendix, when our distance is instead used under the framework of Silva & Tenenbaum (2002), the visualization result shows separation between clusters similar to Figure 1.

Now we discuss related works or compare our framework with the previous works. The seminal t-SNE and UMAP preserve the distance among neighboring data points. It is known that these methods may lose global information, leading to a misunderstanding of the global structure (Coenen & Pearce, 2019; Rudin et al., 2022). To overcome the loss of global information, much effort has been made, for example, by good initialization (McInnes et al., 2018; Kobak & Berens, 2019; Kobak & Linderman, 2021) or by an Euclidean distance preservation between selected non-neighboring points (Fu et al., 2019; Wang et al., 2021). Our proposed GLoMAP improves this global structure preservation by estimating the global distances with adopting a UMAP-like loss function. When it comes to the local distances, both UMAP uses rescale factors that may enable local adaptability of estimated distances. Our local approximation differs from that of UMAP as our local distance estimator is in a closed form and has a consistent property. Additionally, UMAP adopts a fuzzy union to allow their inconsistent coexistence in a single representation of the data manifold, while we use the shortest path search seeking a coherent global metric. Many efforts have been made to extend existing visualization methods to utilize DNNs for generalizability or for handling optimization with lower computational cost, especially when the data size is large (Van Der Maaten, 2009; Gisbrecht et al., 2015; Pai et al., 2019; Sainburg et al., 2021). Our sampling schemes and particle-based algorithm may benefit these approaches in stabilizing the optimization.

The main contributions of this paper are as follows:

(1) We introduce GLoMAP, a novel dimensionality reduction method that captures both global and local structures. The key inventions include locally-adaptive global distances and an annealing-like process for scaling. These innovations enable GLoMAP to produce a *progression from global structure to local details* during optimization, a feature not present in previous methods.

(2) We introduce iGLoMAP, an inductive version of GLoMAP, which avoids the need to re-run the entire algorithm when a new data point is introduced. Our particle-based algorithm leverages the advantages of inductive formulation while maintaining the stability of optimization, similar to transductive learning.

(3) We establish the consistency of our new local geodesic distance estimator, which serves as the building block for the global distance used in both the GLoMAP and iGLoMAP algorithms. The global distance is shown to be an extended metric.

(4) We demonstrate the usefulness of both GLoMAP and iGLoMAP by applying them to simulated and real data and conducting comparative experiments with state-of-the-art methods. All implementations are available at `https://github.com/JungeumKim/iGLoMAP`.

The rest of this article is structured as follows. In Section 2, we provide a brief review of some leading DR methods. For a more complete review, we refer to Rudin et al. (2022). The GLoMAP and iGLoMAP

algorithms are introduced in Section 3. The theoretical analysis and justifications of our distance metric are presented in Section 4. The numerical analysis in Section 5 demonstrates iGLoMAP's competency in handling both global and local information. This article concludes in Section 6.

## 2 Background and the Related Work

In manifold learning, the key design components of DR are: (a) What information to preserve; (b) How to calculate it; (c) How to preserve it. From this perspective, in this section, we review several techniques, such as Isomap, MDS, t-SNE, UMAP in common notation, and discuss their limitations and our improvements. We also discuss PacMAP and PHATE in Section S2.

Let $x_1, \ldots, x_n \in \mathcal{X} \subseteq \mathbb{R}^p$ denote the original dataset. Write $X = [x_1, \ldots, x_n] \in \mathbb{R}^{n \times p}$. Our goal is to find embeddings $z_1, \ldots, z_n \in \mathbb{R}^d$, where $d \ll p$ and $z_i$ is the corresponding representation of $x_i$ in the low-dimensional space. Under many circumstances, $d = 2$ is used for visualization. Write $Z = [z_1, \ldots, z_n] \in \mathbb{R}^{n \times d}$.

### 2.1 Global methods

MDS is a family of algorithms that solves the problem of recovering the original data from a dissimilarity matrix. Dissimilarity between a pair of data points can be defined by a metric, by a monotone function on the metric, or even by a non-metric functions on the pair. For example, in metric MDS, the objective is to minimize *stress*, which is defined by $stress(z_1, ..., z_n) = \left( \sum_{i \neq j} (d_z(z_i, z_j) - f(d_x(x_i, x_j)))^2 \right)^{1/2}$ for a monotonic rescaling function $f$. Depending on how we see the input space geometry, the choices of dissimilarity measures for $d_x$ and $d_z$ would vary. The common choice of the $L_2$ distance corresponds to an implicit assumption that the input data is a linear isometric embedding of a lower-dimensional set into a high-dimensional space (Silva & Tenenbaum, 2002).

Isomap can be seen as a special case of metric MDS where the dissimilarity measure is the geodesic distance, i.e., Isomap tries to preserve the geometry of the data manifold by keeping the pairwise geodesic distances. For this, Isomap first constructs a K-nearest neighbor (KNN) graph where the edges are weighted by the $L_2$ distance among the KNNs. Then it estimates the geodesic distance by applying a shortest path search algorithm, such as Dijkstra's algorithm (Tenenbaum et al., 2000). The crux of Isomap is the use of a shortest path search to approximate global geodesic distances, with the underlying assumption that the manifold is flat and without intrinsic curvature. There have been multiple attempts to improve the geodesic distance estimator under more relaxed conditions. For instance, Isomap has been extended to conformal Isomap (Silva & Tenenbaum, 2002) to accommodate manifolds with curvature. Both their approach and ours construct global distances by finding the shortest path over locally rescaled distances (although we use different rescalers). Our work extends this locally adaptive global approach beyond the MDS framework, addressing the crowding problem identified by Van der Maaten & Hinton (2008). Similarly, other efforts have also sought to handle general manifolds by calibrating local distances. For example, Arias-Castro & Chau (2020) employed the tangential Delaunay complex and Li & Dunson (2019) used a spherelets argument with the decomposition of a local covariance matrix near each data point. In our work, we use a computationally efficient local distance estimator albeit the global distance construction becomes more heuristic. Nevertheless, we believe that these distances from previous work could also be integrated into our framework, providing better theoretical understanding of the chosen global distance, pending improvements in efficient computation.

### 2.2 Local methods

t-SNE, as introduced by Van der Maaten & Hinton (2008), is one of the most cited works in manifold learning for visualization. Instead of equally weighting the difference between the distances on the input and embedding spaces, t-SNE adaptively penalizes this distance gap. More specifically, for each pair $(x_i, x_j)$, $p_{ij}$ defines a relative distance metric in the high-dimensional space, where the relative distance $q_{ij}$ in the low-dimensional space is optimized to match $p_{ij}$ according to the Kullback-Leibler (KL) divergence. For any

two points $x_i$ and $x_j$, define the conditional probability

$$p_{j|i} = \frac{\exp(-\|x_i - x_j\|^2/2\sigma_i^2)}{\sum_{k \neq i} \exp(-\|x_i - x_k\|^2/2\sigma_i^2)}, \tag{1}$$

where $\sigma_i$ is tuned by solving the equation $\log_2 \text{perplexity} = -\sum_j p_{j|i} \log_2 p_{j|i}$. Intuitively, perplexity is a smooth measure of the effective number of neighbors. Define the symmetric probability as $p_{ij} = \frac{p_{i|j} + p_{j|i}}{2n}$. In the embedding space, t-SNE adopts a t-distribution formula for the latent probability, defined by $q_{ij} = (1 + d_{ij}^2)^{-1}/Z$, where $d_{ij} = \|z_i - z_j\|$, and the normalizing constant is $Z = \sum_l \sum_{k \neq l} (1 + d_{kl}^2)^{-1}$. The loss function of t-SNE is the KL divergence between $p$ and $q$, $KL(p\|q) = \sum_{i \neq j} p_{ij} \log \frac{p_{ij}}{q_{ij}}$. The weight matrix of a graph, such as $(p_{ij})$ or $(q_{ij})$, which encodes proximity, is called an affinity matrix. To handle datasets with heterogeneous density, Van Assel et al. (2024) extends t-SNE by maintaining constant entropy for each row in the affinity matrix using an optimal transport framework, defining a doubly stochastic matrix (normalized by both rows and columns) for both input and visualization space.

Another highly cited visualization technique is UMAP (McInnes et al., 2018), which is often considered a new algorithm based on t-SNE (Rudin et al., 2022), known for being faster and more scalable (Becht et al., 2019). UMAP achieves a significant computational improvement by changing the probability paradigm of t-SNE to Bernoulli probability on every single edge between a pair. This removes the need for normalization in t-SNE in both distributions $\{p_{ij}\}$ and $\{q_{ij}\}$ that require summations over the entire dataset. Another innovation brought about by UMAP is a theoretical viewpoint on the local geodesic distance as a rescale of the existing metric on the ambient space (Lemma 1 in McInnes et al. (2018)). McInnes et al. (2018) further develops a local geodesic distance estimator near $x_i$ defined as

$$d_i^{\text{umap}}(x_i, x_j) = \begin{cases} \infty & \text{for } j \notin N_i \text{ or } j = i, \\ (\|x_i - x_j\| - \rho_i)/\hat{\sigma}_i & \text{otherwise,} \end{cases} \tag{2}$$

where $N_i$ is the index set of the KNN of $x_i$, $\rho_i$ is the distance between $x_i$ and its nearest neighbor, and $\hat{\sigma}_i$ is an estimator of the rescale parameter for each $i$. In our work, we address the previously unexplained choices of $\hat{\sigma}_i$ and $\rho_i$ in (2) by proposing a new local geodesic distance estimator and theoretically establishing its consistency.

To embrace the incompatibility between local distances, as indicated by $d_i^{\text{umap}}(x_i, x_j) \neq d_j^{\text{umap}}(x_i, x_j)$, McInnes et al. (2018) treats the distances as *uncertain*, so that those incompatible $d_i^{\text{umap}}(x_i, x_j)$ and $d_j^{\text{umap}}(x_i, x_j)$ may co-exist by a fuzzy union. UMAP defines a weighted graph with the edge weight, or the membership strength, for each edge $(i, j)$ as $\nu_{ij} = p_{ij} + p_{ji} - p_{ij}p_{ji}$, where $p_{ij} = \exp\{-d_i^{\text{umap}}(x_i, x_j)\}$. Meanwhile, on the low-dimensional embedding space, the edge weight for each edge $(i, j)$ is defined by $q_{ij} = \left(1 + a\|z_i - z_j\|^{2b}\right)^{-1}$, where $a$ and $b$ are hyperparameters. Then, UMAP tries to match the membership strength of the representation graph of the embedding space to that of the input space by using a fuzzy set cross entropy, which can be seen as a sum of KL divergences between two Bernoulli distributions such as

$$\mathcal{L}(Z) = \sum_{ij} KL(\nu_{ij}|q_{ij}) = \sum_{ij} \nu_{ij} \log \frac{\nu_{ij}}{q_{ij}} + (1 - \nu_{ij}) \log \frac{(1 - \nu_{ij})}{(1 - q_{ij})}. \tag{3}$$

In UMAP, the global structure is preserved through a good initialization, e.g., through the Laplacian Eigenmaps initialization (McInnes et al., 2018). In our work, we seek a visualization method that does not depend on initialization for global preservation. Another difference of our work is the way to merge the incompatible local distances. We alter the fuzzy union of UMAP by first taking the maximum among finite local distances and then merging the local information by the shortest path search, thereby obtaining global distances.

## 3 Methodology

In this section, we first present a new framework, called GLoMAP, which is a transductive algorithm for nonlinear manifold learning. GLoMAP consists of three primary phases: (1) global metric computation; (2) representation graph construction by information reduction; (3) optimization of the low-dimensional embedding by a stochastic gradient descent algorithm. During the optimization process, the data representation

continuously evolves, initially revealing global structures and subsequently delineating more detailed local structures. Furthermore, we introduce iGLoMAP, an inductive version of GLoMAP, which employs a mapper to replace vector representations of data. iGLoMAP is trained using a proposed particle-based inductive algorithm, designed to mimic the stable optimization process of GLoMAP. Once the mapper is trained, a new data point is easily mapped to its lower dimensional representation without any additional optimization.

## 3.1 Global metric computation

Recall that $\{x_i\}_{i=1}^n$ denotes the original data in $\mathbb{R}^p$ and $\{z_i\}_{i=1}^n$ are the corresponding embeddings in $\mathbb{R}^d$. Our goal is to construct two graphs $G_X$ and $G_Z$ that, respectively, represent the geometry on the input data manifold and the embedding space. We then formulate an objective function as a dissimilarity measure between the graphs $G_X$ and $G_Z$ to project the input space geometry onto the embedding space. The key information to construct the representation graph $G_X$ of the input space is the global distance matrix between *all* pairs of data points. The global distance will be determined based on the local distances, where the locality is defined by the K-nearest neighbor $K_i = \{X_{(k)}(x_i)\}_{k=1}^K$ for each $x_i$. When we obtain the estimate of the local geodesic distance by a rescaled $L_2$ distance, two neighboring points can have two possible rescalers because each of the two points provides its own local view. Therefore, we handle this ambiguity by considering the minimum of the two rescalers, which corresponds to taking the maximum of the two distances. Consequently, the local distance estimate between $x_i$ and $x_j$ is given by

$$\hat{d}_{\text{loc}}(x_i, x_j) = \begin{cases} \frac{\|x_i - x_j\|}{\min\{\hat{\sigma}_{x_i}, \hat{\sigma}_{x_j}\}}, & \text{if } x_j \in K_i \text{ or } x_i \in K_j, \\ \infty, & \text{otherwise,} \end{cases} \tag{4}$$

where $\hat{\sigma}_x$ is the local normalizing estimator defined by

$$\hat{\sigma}_x^2 = \frac{1}{K} \sum_{k=1}^K \|x - X_{(k)}(x)\|^2. \tag{5}$$

The theoretical rationale for the choice of the local normalizing constant $\hat{\sigma}_x^2$ will be provided in Section 4 under the local Euclidean assumption. Using $\hat{d}_{\text{loc}}$ as the local building block, we can construct a global distance matrix. We construct a weighted graph, denoted by $G_{\text{loc}}$, in which $x_i$ and $x_j$ are connected by an edge with the weight $\hat{d}_{\text{loc}}(x_i, x_j)$ if $\hat{d}_{\text{loc}}(x_i, x_j)$ is finite. Given the weighted graph $(G_{\text{loc}}, \hat{d}_{\text{loc}})$, we apply a shortest path search algorithm, e.g., Dijkstra's algorithm, to construct the global distance between any two data points. Note that $\hat{d}_{\text{loc}}(x_i, x_j) < \infty$ when $x_j \in K_i$ or $x_i \in K_j$. Therefore, the shortest path search on $G_{\text{loc}}$ can be seen as an undirected shortest path search on a KNN graph, where the distances are locally rescaled. As a result, for any two data points $x$ and $y$, the search gives

$$\hat{d}_{\text{glo}}(x, y) = \min_P \left( \hat{d}_{\text{loc}}(x, u_1) + \cdots + \hat{d}_{\text{loc}}(u_p, y) \right), \tag{6}$$

where $P$ varies over the graph path between $x$ and $y$. This process is outlined in Algorithm 1. In line 4, $D_2$, sets the distances to zero for all but the $K$-nearest neighbors, indicating disconnections on the neighbor graph. Consequently, the elementwise-max operation in Line 8 yields the intended $\hat{d}_{\text{loc}}$ in equation 4. Note that for the distance of the disconnected elements (nodes), a shortest path search algorithm assigns $\infty$. This implies that when two points are disconnected based on the local graph $G_{\text{loc}}$, they are regarded to be on different disconnected manifolds, so that the global distance is defined as $\infty$.

The search for the shortest path on a neighbor graph with Euclidean distances is well studied by Tenenbaum et al. (2000). However, with our locally adaptive building blocks, comprehending the properties of the resultant global distance becomes challenging. Therefore, the distance construction in equation 6 is heuristic in its nature. In Section 4.2, the construction of the global distance will be explained through the lens of the coequalizer on extended pseudometric spaces. The effectiveness of this approach in capturing the global structure is demonstrated in Examples 1 and 2, as well as in Section 5.

## 3.2 Graph construction and optimization

Having constructed our global distance $\hat{d}_{\text{glo}}$ between all data pairs, we can construct an undirected graph $G_X$ that represents the data manifold. For the embedding space, assuming it is Euclidean with the $L_2$ distance,

---

**Algorithm 1** Global distance construction

---
1: **Input:** the $n \times n$ pairwise $L_2$ distance matrix from $X$ denoted by $D_2$
2: **Fixed input**: the number of neighbor $K$ (default: 15)
3: **Initialization**
4:     Construct the KNN distance matrix $D_K$ from $D_2$
5: **Rescale**
6:     Calculate $n$ distinct $\hat{\sigma}^2$ by a row-wise mean among finite squared distances of $D_K$.
7:     (Rescale) Define $D = D_K[i,j] / \min\{\hat{\sigma}[i], \hat{\sigma}[j]\}$
8:     (Symmetrization) $D_{\mathrm{loc}} = \max\{D, D^T\}$, where max is elementwise-max
9: **Distance construction**
10:     Run $D_{\mathrm{glo}} = \mathcal{S}(D_{\mathrm{loc}})$, where $\mathcal{S}$ is an undirected shortest path search algorithm

---

**Algorithm 2** GLoMAP (Transductive dimensional reduction)

---
1: **Fixed input**: distance matrix $D_{\mathrm{glo}}$, trade-off parameter $\lambda_e$, number of epochs $n_{\mathrm{epoch}}$, learning rate schedule $\alpha_{\mathrm{sch}}$ and tempering schedule $\tau_{\mathrm{sch}}$, $c = 4$.
2: **Learnable input:** randomly initialized representation $Z$
3: **Optimization**
4:     **for** $t = 1, ..., n_{\mathrm{epoch}}$,
5:         Update $\alpha = \alpha_{\mathrm{sch}}[t]$ and $\tau = \tau_{\mathrm{sch}}[t]$
6:         Compute the membership strength $\mu_{ij} = \exp(-D_{\mathrm{glo}}[i,j]/\tau)$, $\mu_{i\cdot} = \sum_j \mu_{ij}$
7:         **for** $k = 1, ..., n_{\mathrm{iter}}$,
8:             Sample mini-batch indices $S = \{i_1, ..., i_m\}$
9:             Sample a neighbor $x_{j_i}$ from $\mu_{j|i}$ for $i \in S$
10:             $\hat{\mathcal{L}}_{\mathrm{glo}} = -\sum_{i \in S} \mu_{i\cdot} \log q_{ij_i} - \lambda_e \sum_{i \in S} \sum_{j \in S}(1 - \mu_{ij}) \log(1 - q_{ij})$
11:             $Z \leftarrow optimizer(Z, \alpha, \nabla_Z \hat{\mathcal{L}}_{\mathrm{glo}}, \mathrm{clip} = c)$
12:         **end for**
13:     **end for**

---

we can construct another undirected graph $G_Z$ representing the embedding space. Specifically, consider the data matrix $X = [x_1, ..., x_n]$ and the embedding matrix $Z = [z_1, ..., z_n]$. The shared index $i \in I = \{1, ..., n\}$ identifies the data point $x_i$ and its embedding $z_i$. Here we construct two weighted graphs of the nodes $I$ denoted by $G_X = (I, E_x)$ and $G_Z = (I, E_z)$. Define the undirected (symmetric) weighted adjacency matrix $E_X$ with a temperature $\tau$ by

$$E_X[i,j] \equiv \mu_{ij} = \exp\{-\hat{d}_{\mathrm{glo}}(x_i, x_j)/\tau\} \tag{7}$$

for $i \neq j$, where the global distance $\hat{d}_{\mathrm{glo}}$ is defined in (6). Note that if the KNN graph $G_{\mathrm{loc}}$ used to construct $\hat{d}_{\mathrm{glo}}$ is not a single connected graph, but several, then there exists $(i,j)$ s.t. $\hat{d}_{\mathrm{glo}} = \infty$. In this case, the weight of the edge $(i,j)$ is 0 by (7). Similarly, with the $L_2$ distance for the embedding graph, define the undirected weighted adjacency matrix $E_Z$ by

$$E_Z[i,j] \equiv q_{ij} = \left(1 + a\|z_i - z_j\|^{2b}\right)^{-1} \tag{8}$$

for $i \neq j$, where $a = 1.57694$ and $b = 0.8951$ are tunable hyperparameters. This definition of $q_{ij}$ and the $a$ and $b$ values were originally used by McInnes et al. (2018).

To force the two graphs $G_X$ and $G_Z$ to be similar, that is, to force the edge weights to be similar, we optimize the sum of KL divergences between the Bernoulli edge probabilities $\mu_{ij}$ and $q_{ij}$. Let $j = 0, ..., n_i$ be the index that $\mu_{ij} > 0$. Then,

$$\mathcal{L}_{\mathrm{glo}}(Z) = \sum_{i,j \in I} KL(\mu_{ij}|q_{ij}) = -\sum_{i=1}^{n}\sum_{j=1}^{n} \mu_{ij} \log q_{ij} - \lambda_e \sum_{i=1}^{n}\sum_{j=1}^{n}(1 - \mu_{ij}) \log(1 - q_{ij}). \tag{9}$$

We call the first term in (9) the positive term because decreasing $-\log q_{ij}$ forces the pair $i, j$ closer, and the second term the negative term because decreasing $-\log(1 - q_{ij})$ works oppositely. We multiply the negative

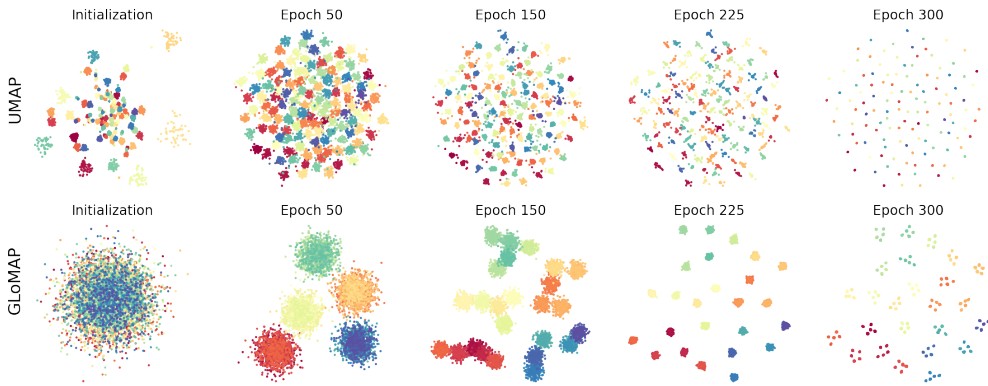

Figure 2: The transductive visualization of the hierarchical dataset (Wang et al., 2021). Top: The results of UMAP. Bottom: From the random initialization, GLoMAP finds first the macro, then meso and then all the micro clusters a progressive way during the optimization.

term by $\lambda_e > 0$ as a tunable weight because in some applications, having control over attractive forces (the positive term) and repulsive forces (the negative term) can be preferred for improved aesthetics (Belkina et al., 2019; Kobak & Berens, 2019). We set $\lambda_e = 1$ as default. This loss construction with the Exponential-and-t formulation is similar to t-SNE and UMAP, where each probability of the pair $(i, j)$ approaches one when the points $(x_i, x_j)$ or $(z_i, z_j)$ are near with each other, and approaches zero when they are far away from each other. We optimize equation 9 through its stochastic version in the following proposition.

**Proposition 1.** *An unbiased estimator of $\mathcal{L}_{\mathrm{glo}}(Z)$ up to a constant multiplication is*

$$\hat{\mathcal{L}}_{\mathrm{glo}}(Z) = -\sum_{i \in S} \mu_{i\cdot} \log q_{ij_i} - \lambda_e \sum_{i \in S} \sum_{j \in S} (1 - \mu_{ij}) \log (1 - q_{ij}), \tag{10}$$

*where $\mu_{i\cdot} = \sum_{j=1}^{n} \mu_{ij}$ and $S$ is a uniformly sampled index set from $I = \{1, ..., n\}$ and $j_i$ is sampled from a conditional distribution $\mu_{j|i} = \frac{\mu_{ij}}{\sum_{j=1}^{n} \mu_{ij}}$.*

**Proof** See Section S1.1.

We apply stochastic gradient descent (SGD) to optimize the loss in (10) with respect to $Z$, decaying the learning rate $\alpha$ and temperature $\tau$. The tempering through decreasing $\tau$ shifts the focus from global to local, thereby catalyzing the progression of visualization from a global to a local perspective in a single optimization process. For a more in-depth discussion of tempering $\tau$, refer to Section S5.1. To stabilize SGD, we adopt two optimization techniques from McInnes et al. (2018). First, the gradient of each summand of $\hat{\mathcal{L}}_{\mathrm{iglo}}(Z)$ is clipped by a fixed constant $c$. Second, the optimizer first updates $Z$ w.r.t. the negative term, then again updates w.r.t. the positive term, which is calculated with the updated $Z$. The described transductive GLoMAP algorithm is in Algorithm 2.

This stochastic optimization approach is highly sought after, especially in our global distance context. The optimization of $\mathcal{L}_{\mathrm{glo}}(Z)$ in (9) is challenging because the number of pairs with nonzero $\mu_{ij}$ can be $O(n^2)$. Note that although the loss formulation $\mathcal{L}_{\mathrm{glo}}(Z)$ in (9) resembles that of t-SNE and UMAP, they do not have the same computational burden because the number of all pairs with non-zero $\mu_{ij}$ is $O(nK) = O(n)$, and thus their schemes consider *all* such pairs at once or sequentially. Therefore, we optimize $\mathcal{L}_{\mathrm{glo}}(Z)$ through $\hat{\mathcal{L}}_{\mathrm{iglo}}(Z)$ in equation 10 with a mini-batch sampling. The caution may arise as each iteration does not necessarily involve the entire dataset. This problem is handled by an inductive formulation in the following section.

**Example 1.** We apply GLoMAP to two exemplary simulated datasets with both global and local structures. The spheres dataset (Moor et al., 2020) has ten inner clusters and an eleventh spread on a large outer shell. Moor et al. (2020) found that UMAP identifies all inner clusters but not the outer shell one. The hierarchical dataset (Wang et al., 2021) features a three-layer tree with five child clusters per parent, across five trees.

Wang et al. (2021) showed UMAP captures micro-level clusters but not meso- or macro-level ones. Detailed data descriptions and hyperparameters are in Section S3 and Section S4 respectively. Figure 1 and 2 show UMAP and iGLoMAP representations at different optimization stages. UMAP uses spectral embedding for initialization, while GLoMAP starts randomly. In Figure 1, GLoMAP on the spheres dataset shows a clear transition from global to local structure during optimization. At epoch 125, it separates the outer shell (purple) from the ten inner clusters, which then become distinct in subsequent epochs. This illustrates how GLoMAP evolves from global to detailed local structures. Similarly, Figure 2 for the hierarchical dataset reveals a progression: by epoch 50, GLoMAP distinguishes all macro-level clusters, then identifies meso-level clusters, and finally, all micro-level clusters, while preserving higher-level structures. We attribute this to GLoMAP's distance metrics that balance global and local information. For the spheres dataset, the distinct separation of the outer shell and inner clusters is due to improved local distance estimation and the use of a smaller rescaler in equation 4, discussed further in Remark 2. The theoretical analysis of our local distance estimate will be provided in Section 4. The effect of the aesthetics parameter $\lambda_e$ is detailed in Section S5.2.

### 3.3 iGLoMAP: The particle-based inductive algorithm

To achieve an inductive dimensional reduction mapping, we parameterize the embedding vector using a mapper $Q_\theta : \mathcal{X} \to \mathbb{R}^d$, where $Q_\theta$ can be modeled by a deep neural network, and the resulted embeddings are $z_i = Q_\theta(x_i)$ for $i = 1, \ldots, n$. Then the loss in equation 9 becomes $\mathcal{L}_{\mathrm{glo}}(Q_\theta(x_1), \ldots, Q_\theta(x_n))$ and is optimized with respect to $Q_\theta$. Due to Proposition 1, we can use an unbiased estimator of $\hat{\mathcal{L}}_{\mathrm{glo}}(Q_\theta(x_1), \ldots, Q_\theta(x_n))$ with $\hat{\mathcal{L}}_{\mathrm{glo}}$ defined in equation 10 for stochastic optimization. We develop a particle-based algorithm to stably optimize the inductive formulation. The idea is that the evaluated particle $z = Q_\theta(x)$ is first updated in a transductive way (as updating $z$ in Algorithm 2), and then $Q_\theta$ is updated accordingly by minimizing the squared error between the original and updated $z$. The proposed particle-based inductive learning is in Algorithm 3. Note that this particle-based approach does not increase any computational cost in handling the DNN compared to a typical deep learning optimization; the DNN is evaluated/differentiated only one time over the entire mini-batch. At the same time, it individually regularizes the gradient of each pair due to the transductive step in Line 12 in Algorithm 3. In our implementation, the optimizer for $\theta$ is set as the Adam optimizer (Kingma & Ba, 2015) with its default learning hyperparameters, i.e., $\beta = (0.9, 0.999)$ with learning rate decay 0.98. The Adam optimizer is renewed every 20 epochs.

In the iGLoMAP algorithm, the entire dataset is affected by every step of stochastic optimization through the mapper $Q_\theta$. This is in contrast to the previous transductive GLoMAP algorithm, where an embedding update of a pair of points does not affect the embeddings of the other points. Due to the generalization property of the mapper, we empirically found that the iGLoMAP algorithm needs fewer iterations than the transductive GLoMAP algorithm, as will be shown in the following example. Note that the particle-based algorithm of iGLoMAP is distinguished from two stage-wise algorithms that first complete the transductive embedding and then train a neural network to learn the embedding (for example, Duque et al. (2020)). In such cases, the transductive embedding stage cannot enjoy the generalization property of the neural network during training.

**Example 2.** We apply iGLoMAP on the spheres and hierarchical datasets in Example 1 with the same hyperparameter settings. As a mapper, we train a basic fully connected neural network as described in Section 5. The visualization results of the training set and also the generalization performance on the test set are presented in Figure 3. In the spheres dataset, the ten inner clusters remain distinct from the outer shell (colored purple), and this separation is still evident in the generalization. For the hierarchical dataset, the test set is generated with different level of random corruption (in noise level, micro level, or meso level) from the training set. As these levels increase, the generalization appears different, yet various levels of clusters can still be observed. We make some other empirical observations. First, as mentioned above, the iGLoMAP algorithm needs fewer iterations compared to the transductive GLoMAP algorithm. We trained it for 150 epochs, in contrast to the 300 epochs for the transductive algorithm. Second, we found that even a small starting $\tau$ does not hinder the discovery of global structures; for an analogous figure to Figure S10, see Figure S29 in Section S8. These findings suggest that the deep neural network might inherently encode some global information. For example, with iGLoMAP on the hierarchical dataset using the default $K = 15$ neighbors, which is generally too few for meso level cluster connectivity, the macro and meso clusters are

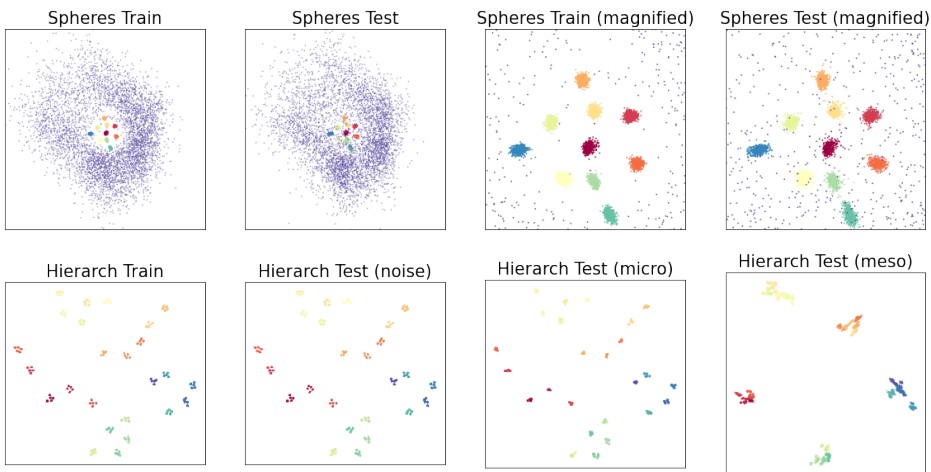

Figure 3: The visualization and generalization performance of iGLoMAP on the hierarchical and spheres dataset. Spheres: All ten inner clusters are identified with the outer shell (purple) scattered around. Hierarchical: all levels of clusters are identified.

---

**Algorithm 3** iGLoMAP (Inductive dimensional reduction)

1: **Fixed input**: distance matrix $D_{\text{glo}}$, trade-off parameter $\lambda_e$, number of epochs $n_{\text{epoch}}$, learning rate schedule $\alpha_{\text{sch}}$ and tempering schedule $\tau_{\text{sch}}$, $c = 4$.
2: **Learnable input:** randomly initialized encoder $Q_\theta$ parametrized by $\theta$
3: **Optimization**
4:     **for** $t = 1, ..., n_{\text{epoch}}$,
5:         Update $\alpha = \alpha_{\text{sch}}[t]$, $\eta = \eta_{\text{sch}}[t]$ and $\tau = \tau_{\text{sch}}[t]$
6:         Compute the membership strength $\mu_{ij} = \exp(-D_{\text{glo}}[i,j]/\tau)$, $\mu_{i\cdot} = \sum_j \mu_{ij}$
7:         **for** $k = 1, ..., \text{n\_iter}$,
8:             Sample mini-batch indices $S = \{i_1, ..., i_m\}$
9:             Sample a neighbor $x_{j_k}$ from $\mu_{j_k | i_k}$ for $i_k \in S$
10:             $Z = \{z_{i_k}, z_{j_k}\}_{k=1}^m$ where $z_{i_k} = Q_\theta(x_{i_k})$, $z_{j_k} = Q_\theta(x_{j_k})$
11:             $\hat{\mathcal{L}}_{\text{glo}} = -\sum_{i_k \in S} \mu_{i_k\cdot} \log q_{i_k j_k} - \lambda_e \sum_{i \in S} \sum_{j \in S} (1 - \mu_{ij}) \log(1 - q_{ij})$
12:             $\tilde{Z} = optimizer(Z, \alpha, \nabla_Z \hat{\mathcal{L}}_{\text{glo}}, \text{clip} = c)$
13:             $\hat{\mathcal{L}}_z = \|Z - \tilde{Z}\|_F^2$ where $\|\cdot\|_F$ is the Frobenius norm
14:             $\theta \leftarrow optimizer(\theta, \eta, \nabla_\theta \hat{\mathcal{L}}_z)$ w.r.t. $\theta$, where $\tilde{Z}$ is regarded as constant
15:         **end for**
16:     **end for**

---

partially preserved even from the start, as shown in Figure S31 in Section S8. Furthermore, only within 5 epochs, all level of clusters are identified. For the sensitivity of $\lambda_e$, see Figure S32 in Section S8. This figure demonstrates $\lambda_e$'s impact, similar to that in the transductive case, but with a much milder effect.

**Remark 1.** *The two main hyperparameters of our method are the negative weight $\lambda_e$ and the tempering schedule of $\tau$. Smaller $\lambda_e$ leads to tighter clustering, while larger values disperse clusters more, as demonstrated in Examples 1 and 2 (Figures S11, S12, and S32 in Section S8). We typically fix $\lambda_e = 1$, and recommend $\lambda_e = 0.1$ for a tighter clustering and $\lambda_e = 10$ for more relaxed cluster shapes. For tempering, since the impact $\tau$'s varies between datasets, we normalize the distances throughout this paper, aiming for a median of 3. Our standard practice begins with $\tau = 1$, reducing to $\tau = 0.1$. This range may not suit every dataset, and thus we provide guidance for adjusting $\tau$. Our method evolves from random noise to global, then local shapes. If it remains noisy for a long time, e.g., over half the epochs, it suggests a too high initial $\tau$, giving insufficient time for local detail development. In such cases, it is advisable to reduce the starting $\tau$; in*

*fact, halving it or even reducing it to a quarter may be beneficial. If meaningful shapes still emerge towards the end, the final $\tau$ might also be too high, indicating the need to halve the final $\tau$.*

## 4 Theoretical Analysis

One of the key innovations of GLoMAP compared with existing methods is the construction of the distance matrix. Roughly speaking, our distance is designed to combine the best of both worlds: like Isomap, the distances are globally meaningful by adopting its shortest path search; like t-SNE and UMAP, the distance is locally adaptive because the shortest path search is conducted over locally adaptive distances. In Section 4.1, we establish that our local distances are consistent geodesic distance estimators under the same assumption utilized in UMAP. Then, in Section 4.2, we provide theoretical insights into our heuristic, which constructs a single global metric space by coherently combining local geodesic estimates. Recall that a geodesic distance $d_M$, given a manifold $M$, is defined by $d_M(x,y) = \inf_s\{\text{length(s)}\}$, where $s$ varies over the set of (piecewise) smooth arcs connecting $x$ to $y$ in $M$.

### 4.1 Local geodesic distance

This section aims to justify the local geodesic distance estimator on a small local Euclidean patch of the manifold $M$. Consider a dataset in $\mathbb{R}^p$, distributed on a d-dimensional Riemannian manifold $M$[1], where $d \leq p$. We make the following assumptions on $(M, g)$ and the data distribution.

   A1. Assume that, for a given point $u \in M$, there exists a geodesically convex neighborhood $U_u \subset M$ such that $(U_u, g)$ composes a $d$-dimensional Euclidean patch with scale $\lambda_u > 0$, i.e., $g = \lambda_u g^*$, where $g^*$ is a Euclidean metric restricted to a $d$-dimensional subspace.

   A2. Assume that $M$ is compact. The data are then assumed to be i.i.d. samples drawn from a uniform distribution on $M$.

A1 characterizes a local neighborhood around a fixed point on $M$ that near the point $u$, $M$ behaves locally as a $d$-dimensional Euclidean space (up to scaling). A2 assumes that the data are uniformly distributed on the manifold. We assume compactness to ensure finite volume, allowing us to define a uniform distribution. Alternatively, we could assume that the manifold $M$ has finite volume under the metric $g$. A2 is beneficial for theoretical reasons, as pointed out in the work of Belkin and Nuyogi on Laplacian eigenmaps (Belkin & Niyogi, 2001; 2003) and also observed by McInnes et al. (2018). Assumptions A1 and A2 together are similar to those of UMAP, which interprets a denser neighborhood as a local manifold approximation with a larger $\lambda_u$ and a sparser neighborhood as one with a smaller $\lambda_u$. Intuitively, this brings an effect of projecting (flattening) the manifold locally onto a Euclidean space. Now we present a distance theorem that connects the local geodesic distance with the $L_2$ distance.

**Theorem 1.** *Denote $x \sim \mathbb{P}$ the uniform probability distribution on $M$ that satisfies A2.*

   (a) *Assume A1 regarding a point $u \in M$ and its open convex neighborhood $U_u$. For any pair $(x, y)$ that belongs to $U_u$ and $\lambda_u > 0$, $d_M(x, y) = \sqrt{\lambda_u} \left\| x - y \right\|_2$.*

   (b) *Assume A1 and A2 regarding a point $u \in M$ with its neighbor $U_u$ and $\lambda_u$. Let $V = vol(M)$. Assume, for an $m \in [0, 1]$, $B_2(u, G_{\mathbb{P},m}(u)) \subset U_u$, where $G_{\mathbb{P},t} : x \in M \mapsto \inf\{r > 0; \mathbb{P}(B_2(x, r)) \geq t\}$ and $B_2(x, r)$ is an $L_2$ ball[2] centered at $x$ with radius $r$. Then, $\sqrt{\lambda_u} = C/\delta_{\mathbb{P},m}(u)$, where $C$ is a universal constant that does not depend on $u$ or $\lambda_u$ but only on $d, m, V$, and*

$$\delta_{\mathbb{P},m} : x \in M \mapsto \sqrt{\frac{1}{m} \int_0^m G_{\mathbb{P},t}^2 dt}, \tag{11}$$

---

[1]Given manifold $M$, a *Riemannian metric* on $M$ is a family of inner products, $\{\langle \cdot, \cdot \rangle_u : u \in M\}$, on each tangent space, $T_u M$, such that $\langle \cdot, \cdot \rangle_u$ depends smoothly on $u \in M$. A smooth manifold with a Riemannian metric is called a *Riemannian manifold*. The Riemannian metric is often denoted by $g = (g_u)_{u \in M}$. Using local coordinates, we often use the notation $g = \sum_{i=1}^d \sum_{j=1}^d g_{ij} dx_i \otimes dx_j$, where $g_{ij}(u) = \langle (\frac{\partial}{\partial x_i})_u, (\frac{\partial}{\partial x_j})_u \rangle_u$.

[2]The ambient metric is used to define $B_2(x, r)$.

*is the distance function to measure (DTM) w.r.t. a distribution $\mathbb{P}$.*

**Proof** See Section S1.2.

Theorem 1 (a) marginally generalizes Lemma 1 from McInnes et al. (2018), which originally introduced the concept of viewing the local geodesic distance as a rescale of the existing metric on the ambient space. In Theorem 1 (b), we connect $\lambda_u$ to DTM (Chazal et al., 2011) by noticing that under the uniform distribution assumption, the probability of a set is closely related to its volume. By this result, for any pair $(x, y)$ in a geodesically convex $U_x$, we have [3]

$$d_M(x, y) \equiv \frac{C}{\delta_{\mathbb{P},m}(x)} \|x - y\|_2. \tag{12}$$

This formulation of $d_M(x, y)$ through DTM $\delta_{\mathbb{P},m}(x)$ is important because its unbiased estimator is identical to $\hat{\sigma}_x$ in equation 5. Therefore, by the result in Theorem 1 (b), we know that $\hat{\sigma}_x/C$ is an unbiased estimator of $1/\sqrt{\lambda_x}$. Now, by using $\hat{\sigma}_x$, we define a local geodesic distance estimator of $d_M(x, y)/C$ as

$$\hat{d}_x(x, y) = \begin{cases} \frac{1}{\hat{\sigma}_x} \|x - y\|, & \text{if } y \in U_x, \\ \infty, & \text{otherwise.} \end{cases} \tag{13}$$

The next theorem establishes the consistency of $\hat{d}_x$.

**Theorem 2.** *Assume A1 and A2 regarding a point $x \in M$ with its neighbor $U_x$ and sample size $n$. Consider a fixed $m \in [0, 1]$ such that $B_2(x, G_{\mathbb{P},m}(x)) \subset U_x$. For any $\xi > 0$, as $n \to \infty$ with with $K$ as the smallest natural number greater than or equal to $mn$,*

$$\mathbb{P}\Big( \exists\, y \in U_x, s.t. \Big| 1 - \frac{d_M^2(x, y)/C^2}{\hat{d}_x^2(x, y)} \Big| \geq \xi \Big) \longrightarrow 0.$$

**Proof** See Section S1.3.

Theorem 2 demonstrates that $\hat{d}_x(x, y)$ converges in probability to $\frac{1}{C} d_M(x, y)$, where $C$ is a universal constant independent of the local center $x$'s choice. Thus, $\hat{d}_x(x, y)$ estimates the geodesic distance up to a constant multiplication. Note that for dimension reduction, the scale of the input space is relatively unimportant, and thus, we regard $C = 1$. Consequently, by establishing an equivalence relationship $d_M(x, y) = \frac{\|x-y\|_2}{\delta_{\mathbb{P},m}(x)}$, we can assert that $\hat{d}_x(x, y)$ consistently estimates $d_M(x, y)$ locally for any $x \in M$.

## 4.2 Construction of global metric space in extended-pseudo-metric spaces

When we apply the local geodesic distance estimation as described in equation 13, treating each data point as the local center, we effectively obtain $n$ different local approximations of the data manifold. In this section, our goal is to coherently glue these approximations together. By treating these as $n$ different metric systems on the shared space $X$ with $|X| = n$, we employ an equivalence relation analogous to Spivak's coequalizer in extended-pseudo-metric spaces. This method presents an alternative to the fuzzy union approach used in McInnes et al. (2018), with both methods aiming to merge local information amidst potential inconsistencies among local assumptions. While the fuzzy union encapsulates the global manifold by aggregating all local distances into a single set, our shortest path approach intricately interweaves these distances to form a unified (extended) metric system.

Let $A$ be the set of all indices of data points $X$, and for all $a \in A$, let $X_a = (X, d_a)$ denote the space defined through the localized metric $d_a$ of UMAP in equation 2 or our approach in equation 13. As shown in McInnes et al. (2018), this localized space $X_a$ for any $a \in A$ is an extended-pseudo-metric space. Metaphorically speaking, the points in $X_a$ that are within a finite distance from $a$ under $d_a$ can be thought of as composing an "island" centered around $a$ (See $X_a$ in Figure S28, the red-colored part). We define $f_a$ as an operator that maps from $X$ to $X_a$. Now, we simply merge these extended-pseudo-metric spaces into one as $X_A = \coprod_{a \in A} X_a$, combining all local information. Let $d_A(y, y') = d_a(y, y')$ if $y, y' \in X_a$ and $d_A(y, y') = \infty$ otherwise. This

---

[3]Note that instead of $\delta_{\mathbb{P},m}(u)$, in (12), we can use $\delta_{\mathbb{P},m}(z)$ for any $z \in U_u$ such that $B_2(z, G_{\mathbb{P},m}(z)) \subset U_u$.

space again is an extended-pseudo-metric space and satisfies the universal property for a coproduct (Spivak (2009), Lemma 2.2). Recalling the island analogy, $X_A$ now contains $n$ islands, as depicted in Figure S28 (in the blue box labeled $X_A$). Note that for $a, a' \in A$, while $X_a$ and $X_{a'}$ are distinct extended-pseudo-metric spaces, they share a very important commonality. They all share the same underlying data points since $X_a = (X, d_a)$ and $X_{a'} = (X, d_{a'})$. Given $x \in X$, however, the two points $x_a = f_a(x) \in X_a$ and $x_{a'} = f_{a'}(x) \in X_{a'}$ are considered distinct points in $X_A$ with $d_A(x_a, x_{a'}) = \infty$. We will sew those inherently-identical points into one in a coherent way through a needle of equivalence relation.

Spivak (2009) considers a coequalizer diagram of sets

$$A \xrightarrow[g_2]{g_1} Z \xrightarrow{q} Y, \tag{14}$$

where $q = [-]$ is the equivalence relation on $Z$ with $x \sim x'$ if there exists $a \in A$ with $x = g_1(a)$ and $x' = g_2(a)$. Given $d_Z$ as a metric on $Z$, define a metric $d_Y$ on $Y$ by

$$d_Y([z], [z']) = \inf(d_Z(p_1, q_1) + d_Z(p_2, q_2) + \cdots + d_Z(p_n, q_n)), \tag{15}$$

where the infimum is taken over all pairs of sequences $(p_1, ..., p_n)$, $(q_1, ..., q_n)$ of elements of $Z$, such that $p_1 \sim z, q_n \sim z'$, and $p_{i+1} \sim q_i$ for all $1 \leq i \leq n-1$. According to Spivak (2009), when $(Z, d_Z)$ is a extended-pseudo-metric space, $(Y, d_Y)$ is another extended-pseudo-metric space and satisfies the universal property of a coequalizer. His construction of coequalizer through equivalence relation can provide in our setting a tool to merge two possibly-conflicting distances into one coherent distance. In the above diagram let $Z = X_A$ and let $g_i$ be the function that maps $X$ to the subset in $X_A$ that is induced by $X_{a_i}$, where we enumerate $A$ through $A = \{a_1, ..., a_n\}$. First, consider $g_1$ and $g_2$. The coequalizer diagram equation 14 regarding $g_1$ and $g_2$ merges the subspaces in $X_A$ that correspond to $X_{a_1}$ and $X_{a_2}$ into a smaller space $(Y, d_Y)$, eliminating the redundancy discussed above. As illustrated in Figure S28, the disconnected two inherently-identical points now are identified as identical, and the two islands are linked, forming a larger island (in the sense that more points are connected with finite distance to others).

Under the design of Spivak (2009) in equation 15, two possibly inconsistent local distances are merged into one by taking the minimum of them. We consider instead to take the maximum between the two as follows. Define a finite max operator (a maximum among finite elements), denoted by

$$\mathrm{fmax}_{a \in A}\{f(a)\} := \begin{cases} \max_{a \in A_f}\{f(a)\}, & \text{if } A_f \neq \emptyset, \\ \infty, & \text{otherwise,} \end{cases} \tag{16}$$

where $A_f = \{a \in A | f(a) < \infty\}$. Recalling that $Z = X_A$, we define the local building block as $d_f(p, q) := \mathrm{fmax}_{x \sim p, y \sim q}\{d_Z(x, y)\}$ to reconcile the incompatibility. Then, the merged distance on $Y$ in equation 15 is replaced as

$$d_Y([z], [z']) = \inf(d_f(p_1, q_1) + d_f(p_2, q_2) + \cdots + d_f(p_n, q_n)). \tag{17}$$

Now, we consider all $g_1, ..., g_n$. We extend the equivalence relation $[-]$, defining $x \sim x'$ if there exist some $a \in A$ and indices $i, j$ such that $g_i(a) = x$ and $g_j(a) = x'$. In this case, $d_Y$ in equation 17 satisfies the conditions of an extended-pseudo-metric.

**Proposition 2.** *The distance $d_Y$ defined in equation 17 is an extended-pseudo-metric, which satisfies: 1) $d_Y(x, y) \geq 0$; 2) $d_Y(x, y) = d_Y(y, x)$; 3) $d_Y(x, y) \leq d_Y(x, w) + d_Y(w, y)$ or $d_Y(x, y) = \infty$.*

**Proof** See Section S1.3.1.

On $Y$, $d_Y$ defines the distances between all pairs in the dataset without inconsistencies among distances. Now, the following theorem says that the global distance of GLoMAP is a special case of the metric $d_Y$ in equation 17, given that two neighboring points are neighbors to each other. Furthermore, it states that in this case, $d_Y$ is an extended metric, which is a stronger notion than an extended pseudo-metric.

**Theorem 3.** *Assume $x \in K_y$ iff $y \in K_x$ for $x, y \in X$, where $K_x$ is the K-nearest neighbor of $x$. Consider $(X_a, d_a)$ constructed by the local geodesic distance estimator defined in equation 13 with $U_x$ approximated by $K_x$. Then $d_Y$ in equation 17 is identical to $\hat{d}_{\mathrm{glo}}$ in equation 6 and an extended metric.*

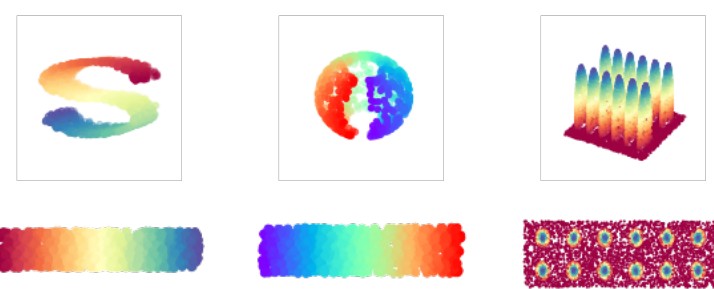

Figure 4: The 3D datasets are generated from the displayed 2D rectangles. Left: S-curve dataset, Middle: Severed Sphere dataset, Right: Eggs dataset.

**Proof** See Section S1.3.2.

**Remark 2.** *If a minimum operator is used instead of the fmax operator, the distance in equation 17 corresponds to the distance defined by Spivak (2009) as in equation 15. The difference between these two distances is subtle, yet the modification in equation 17 reflects our view that when each of two points has a different local scale, the smaller scale should be employed to define the distance between them. In other words, the larger local distance should be used. Conversely, the fuzzy union approach of UMAP results in a final local distance that is shorter than either of the two conflicting local distances.[4] Therefore, our approach and that of McInnes et al. (2018) reflect different philosophical perspectives. We believe that the impact of this philosophical divergence becomes significant in practical applications. For instance, in the Spheres dataset in Examples 1 and 2, points on the outer shell seem very distant from the perspective of the inner small clusters, while from the outer shell's perspective, the inner clusters do not appear comparatively far. We believe that the choice of perspective significantly influences the visualization outcome, as demonstrated in Figure 1, where GLoMAP aligns with the former perspective.*

## 5 Numerical Results

In this section, we compare our approach with existing manifold learning methods and demonstrate its effectiveness in preserving both local and global structures. We consider two scenarios that allow for a fair comparison: 1) cases where lower-dimensional points are smoothly transformed into a high-dimensional space, with the goal being to recover the lower-dimensional points; and 2) scenarios where label information revealing the data structure is available. We provide the GLoMAP and iGLoMAP implementation as a Python package. The detailed learning configurations are in Section S4. Our analysis begins with transductive learning, followed by inductive learning.

### 5.1 Transductive Learning

We compare GLoMAP with existing manifold learning methods, including Isomap, t-SNE, UMAP, PaCMAP, and PHATE, which inherently employ transductive learning. First, we consider a number of three-dimensional datasets to visually compare the DR results, while also quantitatively measuring the performance. Second, we study DR for three high-dimensional datasets that inherently form clusters so that the KNN classification result can be used to measure the DR performance.

---

[4]As described in Section 2.2, the membership strength given by a fuzzy union is defined by $\nu_{ij} = p_{ij} + p_{ji} - p_{ij}p_{ji}$, where $p_{ij} = \exp\{-d_i^{\mathrm{umap}}(x_i, x_j)\}$ and $p_{ji} = \exp\{-d_j^{\mathrm{umap}}(x_i, x_j)\}$. Hence, it can be seen as $\nu_{ij} = \exp\{-x\}$ for some $x \leq \min\{d_i^{\mathrm{umap}}(x_i, x_j), d_j^{\mathrm{umap}}(x_i, x_j)\}$.

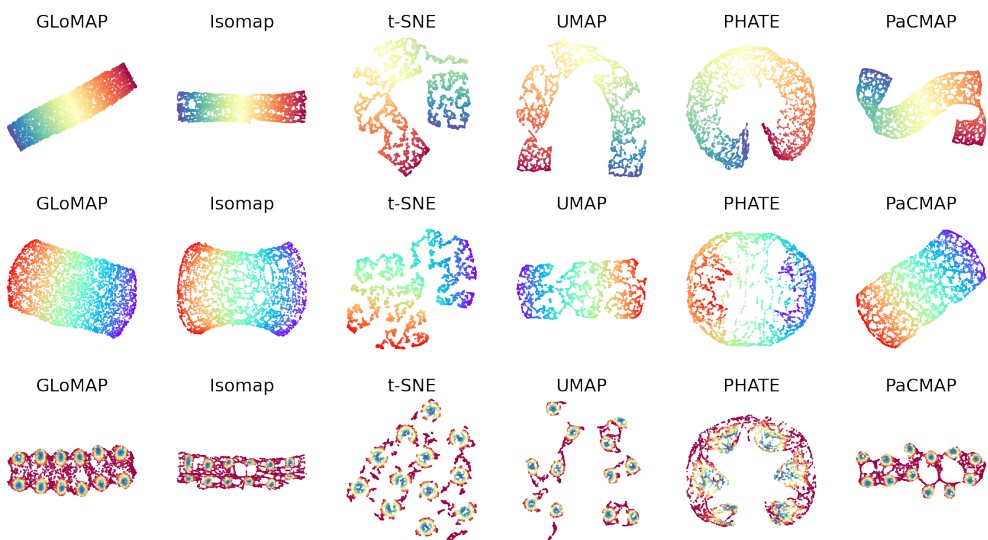

Figure 5: The dimensional reduction for the S-curve, Severed Sphere, Eggs dataset.

### 5.1.1 Manifolds embedded in three dimensional spaces

**Dataset and performance measure.** We study DR for three datasets: S-curve, Severed Sphere, and Eggs, shown in Figure 4. All three datasets are obtained by embedding data points on a two-dimensional box into a three-dimensional space using a smooth function. Detailed data descriptions are presented in Section S3. We adopt two performance measures. The first one is the correlation between the original two-dimensional and the dimensional reduction $L_2$ distances. The second one is the KL-divergence between distance-to-measure (DTM) type distributions, as used in Moor et al. (2020), which is defined by

$$f_\sigma^{\mathcal{X}}(x) := \sum_{y \in \mathcal{X}} \exp\Big(-\frac{dist(x,y)^2}{\sigma}\Big)/\sum_{x \in \mathcal{X}}\sum_{y \in \mathcal{X}} \exp\Big(-\frac{dist(x,y)^2}{\sigma}\Big),$$

where $\sigma > 0$ represents a length scale parameter. For $dist$ in $f_\sigma^{\mathcal{X}}$, we use the $L_2$ distance on the *original* two-dimensional space, and for $dist$ in $f_\sigma^{\mathcal{Z}}$, we use the $L_2$ distance on the two-dimensional embedding space. The denominator is for normalization so that $\sum_{x \in \mathcal{X}} f_\sigma^{\mathcal{X}}(x) = 1$. The KL-divergence given $\sigma$ is

$$KL_\sigma(X, Z) = \sum_i f_\sigma^X(x_i) \log \frac{f_\sigma^X(x_i)}{f_\sigma^Z(z_i)}.$$

We vary $\sigma$ from 0.001 to 10. A larger $\sigma$ focuses more on global preservation, while a smaller $\sigma$ focuses more on local preservation.

**Results.** The visualization results are shown in Figure 5, where GLoMAP demonstrates clear and informative results. Across all three datasets, we observe rectangular shapes and color alignments, indicative of successful preservation of both global and local structures. Isomap and GLoMAP consistently demonstrate visually compatible recovery across all three datasets. PacMAP also achieves this in the Severed Spheres and Eggs datasets. The visualization by PacMAP of the S-curve demonstrates local color alignment although it shows an S-shape. A similar pattern (but with a U-shape) is observed with t-SNE and UMAP in the S-curve. Although the global connectivity of the Eggs dataset is not displayed by t-SNE and UMAP, they demonstrate local structure preservation by local color alignments for each egg shape. Consistent with these visual observations, the quantitative measures, plotted in Figure S30, reveal that GLoMAP has the competitive results overall. Depending on the run, the visualization of GLoMAP of the S-curve and Eggs dataset can be twisted (Figure S36 (c) and (d)). Even when it happens, the performance measures are similar, as the overall global shape is quite similar.

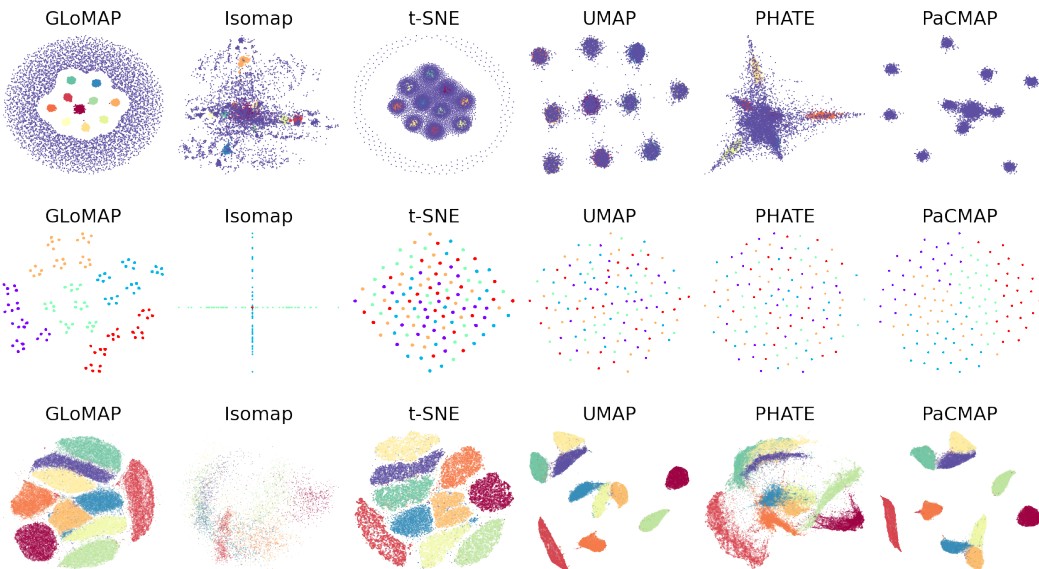

Figure 6: Top: the Spheres. GLoMAP separates points on the larger outer shell from the smaller inner clusters. Middle: the hierarchical dataset. GLoMAP catches all the global, meso and micro level data structures. Bottom: the MNIST. GLoMAP clearly displays the ten digit clusters.

### 5.1.2 High dimensional cluster structured datasets

**Dataset.** We consider the case where the label information revealing the data structure is available. We apply GLoMAP on hierarchical data (Wang et al., 2021), Spheres data (Moor et al., 2020), and MNIST (LeCun et al., 2010). The hierarchical data have five macro clusters, each macro cluster contains five meso clusters, and each meso cluster contains 5 micro clusters. Therefore, the 125 clusters can be seen as 25 meso clusters, where 5 meso clusters also compose one macro cluster. One micro cluster contains 48 observations, and the dimension of the hierarchical dataset is 50. The Spheres data by (Moor et al., 2020) are consisted of 10000 points in 101 dimensional space. Half of the data is uniformly distributed on the surface of a large sphere with radius 25, composing an outer shell. The rest reside inside the shell relatively closer to the origin, composing 10 smaller Spheres of radius 5. Detailed data descriptions are presented in Section S3. The MNIST database is a large database of handwritten digits $0 \sim 9$ that is commonly used to train various image processing systems. The MNIST database contains $70,000$ $28 \times 28$ grey images with the class (label) information, and is available at `http://yann.lecun.com/exdb/mnist/`. The MNIST dataset is regarded as the most important and widely used real dataset for evaluating the data structure preservation of a DR method (Van der Maaten & Hinton, 2008; McInnes et al., 2018; Wang et al., 2021). Since in the later section, iGLoMAP generalizes to unseen data points, we use $60,000$ images for training (and in the later section, the other $10,000$ images for generalization). The baseline methods are also applied to the $60,000$ training images.[5]

**Performance measure.** As a performance measure, we use KNN classification accuracy as described in the followings: Since a data observation comes with a label, we can calculate the classification precision by fitting a k-nearest neighbor (KNN) classifier on the embedding low-dimensional vectors (based on the $L_2$ distance). For the hierarchical dataset, the KNN can measure both local and global preservation; The KNN based on the micro labels shows local preservation, while the KNN based on the macro labels demonstrate global preservation. For the Spheres, KNN is a local measure for the small inner clusters, and a proxy for global separation between the outer shell and the inner clusters. For the MNIST, the KNN measures local preservation. If we assume that the data points within the same class have a stronger proximity than between classes, the KNN classification accuracies on the DR results are a reasonable measure of local preservation.

---

[5]Our computational memory could not handle Isomap on the $60,000$ MNIST images. Therefore, we applied Isomap only on the first $6,000$ MNIST images.

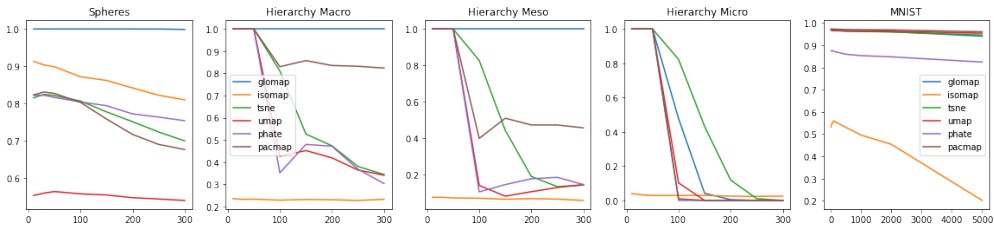

Figure 7: The KNN measures for increasing K.

Note that the labels of the MNIST dataset allow global interpretation by human perceptual understandings of the similarity between numbers. For example, a handwritten 9 is often confused with a handwritten 4.

**Results.** The comparison of the visualization with other leading DR methods is presented in Figure 6 and performance measure in Figure 7. On the Spheres, the proposed GLoMAP shows very intuitive visualization results similar to what one would probably draw on a paper based on the data description. The inner ten clusters are enclosed by the other points that make up a large disk. Similarly, for the hierarchical dataset, we can clearly identify all levels of the hierarchy structure. We can see that although the inner ten clusters are less clearly identified, Isomap also gives the global shape, such that the outer points are well spread out. For methods such as t-SNE, PaCMAP, and UMAP, the outer shell points of the Spheres dataset stick with the inner points making ten clusters. For the hierarchical dataset, no baseline method catches the nested clustering structure; where either the global structure or the meso level clusters is missed. These visual observations are corroborated by the KNN classification plot in Figure 7, which demonstrates the effective KNN classification performance on GLoMAP's DR. For the hierarchical dataset, only GLoMAP shows almost perfect meso and macro level classification. Also, for MNIST, GLoMAP achieves more than 97% of the KNN accuracy numbers, demonstrating GLoMAP's competence in local preservation.

## 5.2 Inductive Learning

We apply iGLoMAP and compare it with other leading parametric visualization methods, including Parametric UMAP (PUMAP) (Pai et al., 2019), TopoAE (Moor et al., 2020), and DIMAL (Sainburg et al., 2021), which can be seen as a parametric Isomap. These methods employ deep neural networks for mapping; they are detailed in further detail in Section S2. For iGLoMAP's mapper, we utilize a fully connected ReLU network with three hidden layers, each of width 128. Following each hidden layer, we incorporate a batch normalization layer and ReLU activation. The network's final hidden layer is transformed into 2 dimensions via a linear layer. For the other methods, we either employ the default networks provided or replicate the network designs used in iGLoMAP. Specifically, for MNIST, PUMAP, and TopoAE are given the configurations (including hyperparameters and network designs) recommended by the original authors.

The DR results are presented in Figure 8.[6] iGLoMAP exhibits similar visual qualities to GLoMAP but with more clearly identifiable global and meso-level clusters for the hierarchical and MNIST datasets. For instance, from the MNIST results in Figure 9, we observe that similar numbers form groups, such as (7,9,4), (0,6), and (8,3,5,2), while the ten distinct local clusters corresponding to the ten-digit numbers are evident. The generalization capability of iGLoMAP is depicted in Figure 9 for MNIST and in Figure S33 in Section S8 for the S-curve, Severed, and Eggs datasets, demonstrating almost identical visualizations for unseen data points. Interestingly, the enhanced performance of parametric UMAP over (transductive) UMAP, despite sharing the same framework, reinforces our conjecture in Example 2 that incorporating DNNs aids in preserving global information. Nonetheless, PUMAP's performance on the spheres dataset serves as a cautionary example, indicating that the application of DNNs is not a universal solution for bridging the gap in global information representation. On the Eggs dataset, PUMAP often displayed a more pronounced disconnection than that shown in Figure 8, as shown in Figure S36 (a) in Section S8. We also observed that depending on the specific run, the Eggs and S-curve visualizations of iGLoMAP can appear twisted, as in Figure S36 (b) in Section S8. The numerical performance metrics are presented in Figures S34 and S35 in

---

[6]DIMAL collapsed on the hierarchical dataset and is thus not included in Figure 8.

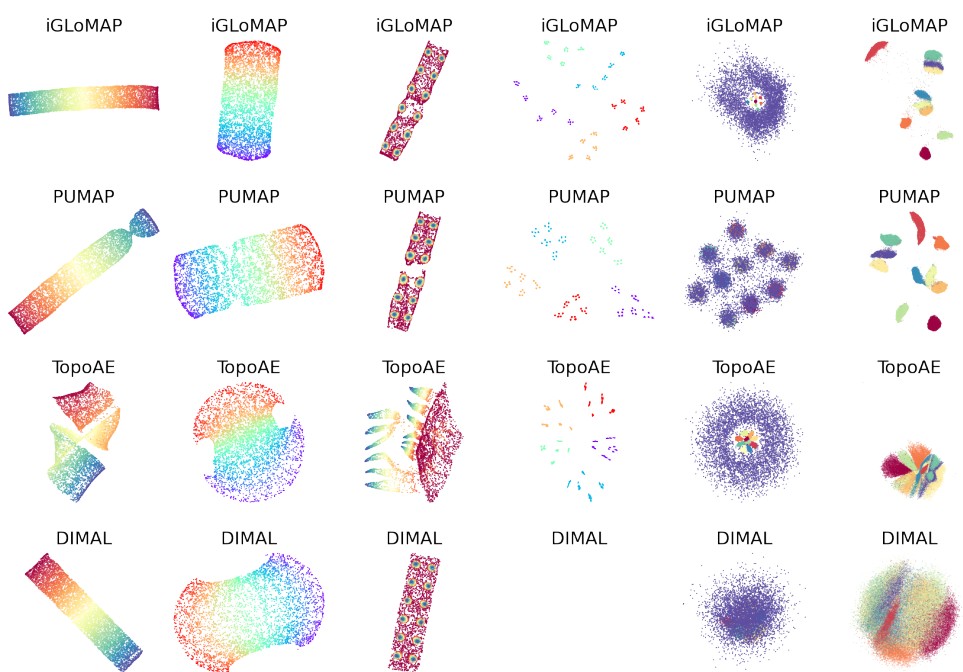

Figure 8: Comparison with other inductive models that utilize deep neural network.

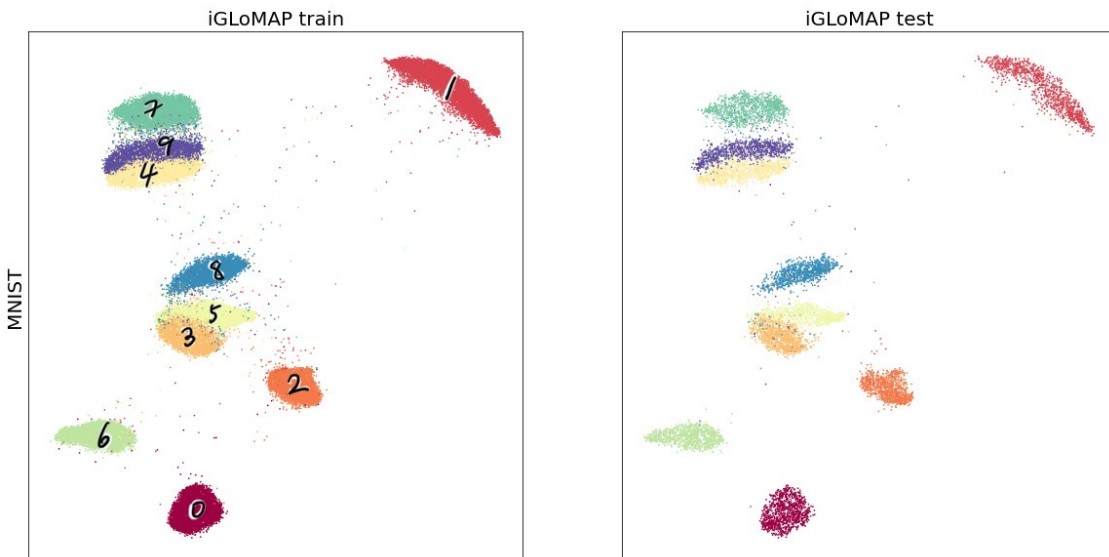

Figure 9: The iGLoMAP generalizations for the MNIST. The generalization on the test set is almost identical to the original DR on the training set.

Section S8, demonstrating that iGLoMAP is compatible with or outperforms other methods on the datasets used.

## 6 Conclusion

In this paper, we proposed GLoMAP, a unified framework for high-dimensional data visualization capable of both global and local preservation. By tempering $\tau$, we observed a transition from global formation to local

detail within a single optimization process. This is attributed to the global distance construction with locally adaptive distances as the building blocks, offering an alternative to UMAP's fuzzy union. Furthermore, our algorithms, which are randomly initialized, do not rely on optimal initialization for global preservation. Additionally, we extend the GLoMAP algorithm to its inductive variant, iGLoMAP, by incorporating deep learning techniques to learn a dimensionality reduction mapping.

We now mention several future research directions on GLoMAP and iGLoMAP. First, the full impact of using DNNs remains to be explored. We have observed that the use of DNNs can encode some global information as shown with iGLoMAP on hierarchical dataset with small $K$ (Example 2 and Figure S31) and parametric UMAP on various datasets as well (Figure 8). Also, in Section 5, the output of iGLoMAP is generally similar to that of GLoMAP, but not exactly the same. More detailed investigation seems necessary. Additionally, reducing the computational demands of our algorithm is a priority. While we have reduced the computational cost for local distance estimation, managing the global distances, conversely, increases the cost. The shortest path search step represents the most computationally intensive aspect of our framework. Depending on the algorithm, the shortest path search can cost up to $O(n^3 K)$. In our experiment, when the number of points is more than 60,000, we have experienced a significant computational bottleneck. One possible resolution could be a landmark extension such as that of t-SNE or Isomap (Van der Maaten & Hinton, 2008; Silva & Tenenbaum, 2002). Moreover, the neighbor sampling scheme, which involves summing membership scores over $n$ distances when all data points are connected, incurs costs comparable to the normalization of t-SNE. Currently, this issue can be addressed through approximations, as discussed in Section S5.3, but further development to improve computational efficiency would be advantageous. Currently, our implementation is solely in Python, and it could benefit from optimization through advanced languages and interfaces like Numba, Cython, or C++. See Figure S20 for a comparison of the current computational time with other leading baselines. In this paper, we did not consider the problem of noise in data. Robustness against noise is an important problem of dimensional reduction, which is worth a subsequent study, but is beyond the scope of current work. Additionally, the uniform assumption applied in this study may be too rigid. Finally, the current applicability of our proposed methods is limited to datasets without missing data. Determining how to effectively incorporate missing data into the algorithm remains an intriguing and challenging area for future research. We believe that our results nevertheless serve as a valuable addition towards visualization of both global and local structures in data, useful in practice.

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

# Appendix

## S1   Proofs of Theorems and Propositions

In this Appendix we prove the following theorem from Section 3 and Section 4.

### S1.1   Proof of Proposition 1

We only need to show that $\mathcal{L}_{\mathrm{glo}}(Z) = c\mathbb{E}_S[\hat{\mathcal{L}}_{\mathrm{iglo}}(Z)]$ for some constant $c > 0$, where

$$\mathcal{L}_{\mathrm{glo}}(Z) = -\sum_{i=1}^{n}\sum_{j=1}^{n}\mu_{ij}\log q_{ij} - \lambda_e\sum_{i=1}^{n}\sum_{j=1}^{n}(1-\mu_{ij})\log(1-q_{ij}). \tag{S2}$$

We first rewrite the first term in (S2) by using an expectation form. We say $i \sim \mu_{i\cdot}$ when $i$ follows a discrete distribution over $\{1, ..., n\}$ with probability $(\mu_{1\cdot}, ..., \mu_{n\cdot})$. Likewise, we say $j|i \sim \mu_{j|i}$ when $j|i$ follows a discrete distribution over $\{1, ..., n\}$ with probability $(\mu_{1|i}, ..., \mu_{n|i})$ where $\mu_{j|i} := \frac{\mu_{ij}}{\sum_j \mu_{ij}}$. Then we can rewrite (S2) as

$$\sum_{i=1}^{n}\sum_{j=1}^{n}\mu_{ij}\log q_{ij} = \mathbb{E}_i\mathbb{E}_{j|i}\big[\log q_{ij}\big],$$

where $j|i \sim \mu_{j|i}$ and $i \sim \mu_{i\cdot}$. Now, for easy sampling, we further consider $U$, a uniform distribution over $\{1, ..., n\}$. Then, now we have a formulation for importance sampling

$$\mathbb{E}_i\mathbb{E}_{j|i}\big[\log q_{ij}\big] = \mathbb{E}_{i \sim U}\Big[\frac{\mu_{i\cdot}}{1/n}\mathbb{E}_{j|i}\big[\log q_{ij}\big]\Big] = n\mathbb{E}_{i \sim U}\Big[\mu_{i\cdot}\mathbb{E}_{j|i}\big[\log q_{ij}\big]\Big].$$

Its one-sample unbiased estimator is $n\mu_{i\cdot}\log q_{ij_i}$ where $i \sim U$ and $j_i \sim \mu_{j|i}$. Now, $S$ is simply a multi-sample version of the one-sample case. Therefore, $\sum_{i \in S}\mu_{i\cdot}\log q_{ij_i}$ is an unbiased estimator of $\sum_{i=1}^{n}\mu_{i\cdot}\log q_{ij_i}$ up to a constant multiplication, i.e., $c = n/|S|$.

Now, rewrite the second term in (S2) again by using an expectation form. This time, however, we use two independent uniform distributions $i \sim U$ and $j \sim U$. That is,

$$\sum_{i=1}^{n}\sum_{j=1}^{n}(1-\mu_{ij})\log(1-q_{ij}) = n^2\mathbb{E}_{i \sim U}\mathbb{E}_{j \sim U}\big[(1-\mu_{ij})\log(1-q_{ij})\big]. \tag{S3}$$

When we have two independent copies $i \sim U$ and $j \sim U$, an unbiased estimator of (S3) is $n^2(1-\mu_{ij})\log(1-q_{ij})$. To extend this idea, when we have $i \perp \{j_1, ..., j_{m-1}\}$ where $j_k$'s are i.i.d. from $U$, again, an unbiased estimator of (S3) is

$$\frac{n^2}{m-1}\sum_{k=1}^{m-1}(1-\mu_{ij_k})\log(1-q_{ij_k}).$$

Note that $S = \{i_1, ..., i_m\}$ is a set of independent draws from $U$. Therefore, for any $i_k$, we have independency $i_k \perp \{i_1, .., i_{k-1}, i_{k+1}, ..., i_m\}$. Therefore, we have another unbiased estimator of (S2), that is

$$\frac{n^2}{m(m-1)}\sum_{i \in S}\sum_{j \in S\setminus i}(1-\mu_{ij})\log(1-q_{ij}).$$

Note that $q_{ii} = 0$ so that $\log(1-q_{ii}) = 0$. Therefore, above estimator is simplified without any change in value to

$$\frac{n^2}{m(m-1)}\sum_{i \in S}\sum_{j \in S}(1-\mu_{ij})\log(1-q_{ij}).$$

This proves that $\sum_{i \in S}\sum_{j \in S}(1-\mu_{ij})\log(1-q_{ij})$ is an unbiased estimator of $\sum_{i=1}^{n}\sum_{j=1}^{n}(1-\mu_{ij})\log(1-q_{ij})$ up to a constant multiplication, i.e., $c = n^2/m(m-1)$.

Note that, for the two terms, the constant multiplications applied for unbiasedness are different. However, because $\lambda_e$ is a flexible tuning hyperparameter, we can simply redefine $\lambda_e$ by multiplying the ratio of the two constant multiplications. ∎

### S1.2 Proof of Theorem 1

**Proof** [Proof of (a)] Denote the Euclidean metric on $\mathbb{R}^p$ by $\bar{g}$ and that of scale $\lambda_u$ as $\lambda_u \bar{g}$. Since $U_u$ is Euclidean and convex, the shortest geodesic on $U_u$ that connects any $x, y \in U_u$ is a straight line, that is a straight line in the Euclidean geometry. We define

$$\Gamma_M = \{\gamma | \gamma(0) = x, \gamma(1) = y, \text{smooth curve on } M\},$$
$$\Gamma_U = \{\gamma | \gamma(0) = x, \gamma(1) = y, \text{smooth curve on } U_u\}.$$

Then, for any $x, y \in U_u$,

$$d_M(x, y) = \inf_{\gamma \in \Gamma_M} \{length(\gamma)\} = \inf_{\gamma \in \Gamma_U} \{length(\gamma)\} = \inf_{\gamma \in \Gamma_U} \int_0^1 \sqrt{g(\dot{\gamma}(t), \dot{\gamma}(t))} dt$$
$$= \inf_{\gamma \in \Gamma_U} \int_0^1 \sqrt{\lambda_u \bar{g}(\dot{\gamma}(t), \dot{\gamma}(t))} dt = \sqrt{\lambda_u} \inf_{\gamma \in \Gamma_U} \int_0^1 \|\dot{\gamma}(t)\| dt = \sqrt{\lambda_u} d_2(x, y).$$

In the second equation, we changed the infimum over $\Gamma_M$ to $\Gamma_U$. This is because by the geodesic convexity of $U_u$, the shortest geodesic arc that connects $x, y$ is in $\Gamma_U$ (which is the shortest arc that connects $x, y$.) ∎

**Proof** [Proof of (b)] Under $A2$, $X_1 \sim Unif(M)$, i.e., pdf $f(x) = \frac{1}{V} 1_{x \in M}$, where $V = Vol(M)$. For $m \in [0, 1]$, define the global constants $C_d := \frac{2\pi^{d/2}}{\Gamma(\frac{d}{2}+1)}$ and $C_{d,m,V} := \left(\sqrt{\frac{d+2}{d}} \left(\frac{C_d}{Vm}\right)^{\frac{1}{d}}\right)^{-1}$. Assume $B_2(x, G_{\mathbb{P},m}) \in U_u$ on which $g = \lambda_u \bar{g}$. Since $U_u$ is $d$-dimensional Euclidean, we can consider new coordinates $x_1, ..., x_d$ as spanning $U_u$ so that the volume element is $dV = \sqrt{det(g)} dx^1 \wedge \cdots \wedge dx^d$. Then, the volume of $Vol(B_2(x, t))$ is

$$Vol(B_2(x, t)) = \int_{B_2(x,t)} \sqrt{det(g)} dx^1 \wedge \cdots \wedge dx^d$$
$$= \lambda_u^{\frac{d}{2}} \int_{B_2(x,t)} dx^1 \wedge \cdots \wedge dx^d$$
$$= \lambda_u^{\frac{d}{2}} \frac{2\pi^{d/2}}{\Gamma(\frac{d}{2}+1)} t^d$$
$$= \lambda_u^{\frac{d}{2}} t^d C_d$$

Therefore, for $B_2(x, t) \in U_u$,

$$\mathbb{P}(B_2(x, t)) = \frac{Vol(B_2(x, t))}{V} = \frac{\lambda_u^{\frac{d}{2}} t^d C_d}{V}.$$

In our setting,

$$\mathbb{P}(B_2(x, t)) \geq u \Leftrightarrow t \geq \frac{1}{\sqrt{\lambda}} \left(\frac{uV}{C_d}\right)^{\frac{1}{d}},$$

and therefore,

$$G_{\mathbb{P},u} = \frac{1}{\sqrt{\lambda}} \left(\frac{uV}{C_d}\right)^{\frac{1}{d}}.$$

Therefore, the DTM is

$$
\begin{aligned}
\delta^2_{\mathbb{P},m}(x) &= \frac{1}{m}\int_0^m \left(\frac{1}{\sqrt{\lambda}}\left(\frac{uV}{C_d}\right)^{\frac{1}{d}}\right)^2 du \\
&= \frac{1}{m}\frac{1}{\lambda}\left(\frac{V}{C_d}\right)^{\frac{2}{d}}\int_0^m u^{\frac{2}{d}}\,du \\
&= \frac{d}{d+2}\frac{1}{m}\frac{1}{\lambda}\left(\frac{V}{C_d}\right)^{\frac{2}{d}} m^{1+\frac{2}{d}} \\
&= \frac{d}{d+2}\frac{1}{\lambda}\left(\frac{V}{C_d}\right)^{\frac{2}{d}} m^{\frac{2}{d}}.
\end{aligned}
$$

Therefore,

$$
\frac{1}{\sqrt{\lambda_u}} = \delta_{\mathbb{P},m}(x)\sqrt{\frac{d+2}{d}}\left(\frac{C_d}{Vm}\right)^{\frac{1}{d}} = \frac{\delta_{\mathbb{P},m}(x)}{C_{d,m,V}}.
$$

$\blacksquare$

### S1.3 Proof of Theorem 2

**Proof**. In this proof, we make use of the limiting distribution theorem regarding the empirical DTM in Chazal et al. (2017). First, we restate the theorem.

**Theorem 4** (Theorem 5 in Chazal et al. (2017)). *Let $\mathbb{P}$ be some distribution in $\mathbb{R}^d$. For some fixed $x$, assume that $F_x$ is differentiable at $F_x^{-1}(m)$, for $m \in (0,1)$, with positive derivative $F_x'(F_x^{-1}(m))$. Define $\hat{\delta}_{\mathbb{P}_n,m}(x) = \sum_{k=1}^K \|x - X_{(k)}(x)\|^2/K$ for $K = \lceil * \rceil mn$. Then we have*

$$
\sqrt{n}(\hat{\delta}^2_{\mathbb{P}_n,m}(x) - \delta_{\mathbb{P},m}(x)) \xrightarrow{d} N(0, \sigma_x^2),
$$

*where $\sigma_x^2 = \frac{1}{m^2}\int_0^{F_x^{-1}(m)}\int_0^{F_x^{-1}(m)}[F_x(s \wedge t) - F_x(s)F_x(t)]\,ds\,dt$.*

Now, assume A1 and A2 regarding a point $x \in M$. From A2, we have

$$
d_M(x,y) \equiv \frac{\|x-y\|_2 C_{d,m,V}}{\delta_{\mathbb{P},m}(x)}.
$$

$$
\left|1 - \frac{d_M^2(x,y)}{C_{d,m,V}^2 \hat{d}_x^2(x,y)}\right| \geq \epsilon
$$
$$
\Leftrightarrow \left|1 - \frac{\hat{\sigma}_x^2}{\delta_{\mathbb{P},m}^2(x)}\right| \geq \epsilon
$$
$$
\Leftrightarrow \left|\delta_{\mathbb{P},m}(x)^2 - \hat{\sigma}_x^2\right| \geq \epsilon \delta_{\mathbb{P},m}^2(x)
$$

Here, $\epsilon_x := \epsilon\delta_{\mathbb{P},m}(x)^2$ is a fixed nonnegative number. Therefore,

$$
\mathbb{P}\left(\exists y \in U_x, s.t. \left|1 - \frac{1}{C_{d,m,V}^2}\frac{d_M^2(x,y)}{\hat{d}_x^2(x,y)}\right| \geq \epsilon\right) = \mathbb{P}\left(\left|\delta_{\mathbb{P},m}^2(x) - \hat{\sigma}_x^2\right| \geq \epsilon_x\right)
$$

By Theorem 4, when $F_x$ is differentiable at $F_x^{-1}(m)$ for $m \in (0,1)$, with positive derivative $F_x'(F_x^{-1}(m))$, where $F_x(t) = \mathbb{P}\left(\left\|X - x\right\|^2 \leq t\right)$,

$$
\sqrt{n}(\delta_{\mathbb{P},m}^2(x) - \hat{\sigma}_x^2) \xrightarrow{d} N(0, \eta_x^2),
$$

for some fixed $\eta_x^2$. We first check if the condition satisfies our setting. Note that by the definition of $G_{\mathbb{P},m}(x)$, we know that $\mathbb{P}(B_2(x, G_{\mathbb{P},m}(x))) \geq m$, i.e., $F_x(G_{\mathbb{P},m}(x)) \geq m$. Since $F_x(t)$ is a non-decreasing function of $t$, let us consider $t \leq G_{\mathbb{P},m}(x)$. By the proof of Theorem 1, we know that

$$F_x(t) = \mathbb{P}\left(\left\|X - x\right\|^2 \leq t\right) = \mathbb{P}\left(B_2(x, \sqrt{t})\right) = ct^{d/2},$$

for some constant $c > 0$. Therefore, $F_x(t)$ is differentiable for $t \leq G_{\mathbb{P},m}(x)$, and its derivative is always positive as long as $t > 0$. Note that $F_x^{-1}(m) > 0$ when $m \in (0,1)$. Therefore, we can apply Theorem 4 in our setting, so that $\delta_{\mathbb{P},m}^2(x) - \hat{\sigma}_x^2 = op(1)$. Therefore,

$$\mathbb{P}\left(\left|\delta_{\mathbb{P},m}^2(x) - \hat{\sigma}_x^2\right| \geq \epsilon_x\right) \to 0.$$

■

### S1.3.1 Proof of Proposition 2

Define for $x, y \in Y$, $\mathcal{A}(x, y)$ as a set of all sequences $\{p_i, q_i\}_{i=1}^n$ of elements of $X_A$, that satisfy $p_{i+1} \sim q_i$ for all $1 \leq i \leq n - 1$ and $[p_1] = x$, $[q_n] = y$.

1. Since each $d_f$ is non-negative, by definition $d_Y$ is non-negative.

2. For a sequence in $\mathcal{A}(x, y)$, its reverse exists and is also a sequence in $\mathcal{A}(y, x)$. Now, the symmetry of $d_f$ makes the path distance of any ordered sequence $\{p_i, q_i\}_{i=1}^n \in \mathcal{A}(x, y)$ the same as that of the reversed sequence $\{\tilde{p}_i, \tilde{q}_i\}_{i=1}^n \in \mathcal{A}(y, x)$, where $p_j = \tilde{q}_{n+1-j}$ and $q_j = \tilde{p}_{n+1-j}$. In other words,

$$d_f(p_1, q_1) + \cdots + d_f(p_n, q_n) = d_f(\tilde{p}_1, \tilde{q}_1) + \cdots + d_f(\tilde{p}_n, \tilde{q}_n).$$

Therefore, we have $d_Y(x, y) = d_Y(y, x)$ because

$$\inf_{\mathcal{A}(x,y)} d_f(p_1, q_1) + \cdots + d_f(p_n, q_n) = \inf_{\mathcal{A}(y,x)} d_f(\tilde{p}_1, \tilde{q}_1) + \cdots + d_f(\tilde{p}_n, \tilde{q}_n).$$

3. Let $d_Y(x, y) < \infty$. Choose any $w \in Y$. If any of $d_Y(x, w)$ or $d_Y(w, y)$ is $\infty$, then the inequality is true. Now, consider both $d_Y(x, w)$ and $d_Y(w, y)$ to be finite. Then, there exist minimizing sequences $\{\hat{p}_i, \hat{q}_i\}_{i=1}^n \in \mathcal{A}(x, w)$ and $\{\tilde{p}_i, \tilde{q}_i\}_{i=1}^n \in \mathcal{A}(w, y)$ such that $d_Y(x, w) = d_f(\hat{p}_1, \hat{q}_1) + \cdots d_f(\hat{p}_n, \hat{q}_n)$ and $d_Y(w, y) = d_f(\tilde{p}_1, \tilde{q}_1) + \cdots d_f(\tilde{p}_n, \tilde{q}_n)$ with $[\hat{q}_n] = [\tilde{p}_1] = w$. Note that when we merge the two paths into one, we have a path connecting $x$ and $y$. Now, similarly to $\mathcal{A}(x, y)$, define $\mathcal{A}_2(x, y)$ as a set of all sequences $\{p_i, q_i\}_{i=1}^{2n}$ of elements of $X_A$, that satisfy $p_{i+1} \sim q_i$ for all $1 \leq i \leq n - 1$ and $p_1 \sim x$, $q_{2n} \sim y$.

$$
\begin{aligned}
d_Y(x, y) :&= \inf_{\mathcal{A}(x,y)} \left(d_f(p_1, q_1) + \cdots + d_f(p_n, q_n)\right) \\
&= \inf_{\mathcal{A}_2(x,y)} \left(d_f(p_1, q_1) + \cdots + d_f(p_{2n}, q_{2n})\right) \\
&\leq d_f(\hat{p}_1, \hat{q}_1) + \cdots d_f(\hat{p}_n, \hat{q}_n) + d_f(\tilde{p}_1, \tilde{q}_1) + \cdots + d_f(\tilde{p}_n, \tilde{q}_n) \\
&= d_Y(x, w) + d_Y(w, y).
\end{aligned}
$$

The second equality is because the shortest distance can always be achieved by a path of length $n$ since $card(X) = card(Y) = n$.

■

### S1.3.2 Proof of Theorem 3

We have $g_1, ..., g_n$, and each $x \in X$ is mapped to a distinct point $g_i(x) \in X_A$, i.e., $g_i(x) \neq g_j(x)$ if $i \neq j$. Therefore, for each $p \in X_A$, we can uniquely identify the index $i$ and $x \in X$ s.t. $p = g_i(x)$, i.e., $x = g_i^{-1}(p)$. Since the image of $g_i$ does not overlap i.e., $g_i(X) \cap g_j(X) = \emptyset$ for $i \neq j$, we can define $g^{-1}$, which is a unified

function of all $g_1^{-1}, ..., g_n^{-1}$, i.e., for $p \in X_A$, $g^{-1}(p) := g_i^{-1}(p)$ for $i$ such that $p \in g_i(X)$. Now, for $p, q \in X_A$, we have $d_f(p, q) < \infty$ only when $g^{-1}(p) \in K_{g^{-1}(q)}$ or $g^{-1}(q) \in K_{g^{-1}(p)}$. By the assumption of Theorem 3, when $g^{-1}(p) \in K_{g^{-1}(q)}$, we always have $g^{-1}(q) \in K_{g^{-1}(p)}$. This ensures that when $d_{g^{-1}(p)}(g^{-1}(p), g^{-1}(q)) < \infty$, we have $d_{g^{-1}(q)}(g^{-1}(p), g^{-1}(q)) < \infty$. Therefore, in this case,

$$
\begin{aligned}
d_f(p, q) &= \max\{d_{g^{-1}(p)}(g^{-1}(p), g^{-1}(q)), d_{g^{-1}(q)}(g^{-1}(p), g^{-1}(q))\} \\
&= \hat{d}_{\mathrm{loc}}(g^{-1}(p), g^{-1}(q)),
\end{aligned}
$$

for $\hat{d}_{\mathrm{loc}}$ defined in equation 4. Thus, for $x, y \in Y$, we have

$$
\begin{aligned}
d_Y(x, y) &:= \inf_{\mathcal{A}(x,y)} (d_f(p_1, q_1) + d_f(p_2, q_2) + \cdots + d_f(p_n, q_n)) \\
&= \inf_{\mathcal{A}(x,y)} (d_{\mathrm{loc}}(g^{-1}(p_1), g^{-1}(q_1)) + \cdots + d_{\mathrm{loc}}(g^{-1}(p_n), g^{-1}(q_n))) \\
&= \min_{P(\tilde{x}, \tilde{y})} (\hat{d}_{\mathrm{loc}}(\tilde{x}, u_1) + \cdots + \hat{d}_{\mathrm{loc}}(u_p, \tilde{y})) = \hat{d}_{\mathrm{glo}}(\tilde{x}, \tilde{y})
\end{aligned}
$$

with $\hat{d}_{\mathrm{glo}}(x, y)$ defined in equation 6, where $\tilde{x} = g^{-1}(p_1) \in X$ and $\tilde{y} = g^{-1}(q_n) \in X$. The second equality is due to $g^{-1}(p) = g^{-1}(q)$ for $p \sim q$. Note that the $x, y \in Y$ are identified by $\tilde{x}, \tilde{y} \in X$, so with a slight abuse of notation, we can say $d_Y(x, y) = \hat{d}_{\mathrm{glo}}(x, y)$.

Now, we show that $d_Y$ in this case is an extended-metric. Given Proposition 2, we only need to show that $d_Y(x, y) = 0$ iff $x = y$. Since $d_a$ satisfies $d_a(x, y) = 0$ iff $x = y$, we have $d_f(p, q) = 0$ iff $p = q$. Given this, consider $x, y \in Y$. It is obvious that $d_Y(x, y) = 0$ if $x = y$; we can take $p_i$ and $q_i$ so that $p_i = q_i$ and $[p_i] = [q_i] = x$ for $i = 1, ..., n$, which gives $0 \le d_Y(x, y) \le d_f(p_1, q_1) + d_f(p_2, q_2) + \cdots + d_f(p_n, q_n) = 0$. If $d_Y(x, y) = 0$, there must exist a sequence $\{p_i, q_i\}_{i=1}^n \in \mathcal{A}(x, y)$ such that $d_f(p_1, q_1) + d_f(p_2, q_2) + \cdots + d_f(p_n, q_n) = 0$. Since if $p_i \ne q_i$, then $d_f(p_i, q_i) > 0$, we have $p_i = q_i$ for $i = 1, ..., n$. Since $p_{i+1} \sim q_i$ by the definition of $\mathcal{A}(x, y)$ and since $[q_i] = [p_i]$, we conclude that $x = [p_1] = \cdots = [p_n] = y$. ∎

## S2   Other Related Work

In this section, we discuss additional related work.

**PaCMAP**   (Wang et al., 2021) propose PaCMAP to preserve both the local and global structure. The loss of PaCMAP is derived from an observational study of state-of-the-art dimension reduction methods such as t-SNE (Van der Maaten & Hinton, 2008) and UMAP (McInnes et al., 2018). The global preservation of PaCMAP is based on initialization through PCA and some optimization scheduling techniques. Our development of global distances may be adopted by PacMAP instead of its current use of Euclidean distance.

**PHATE**   (Moon et al., 2017) propose PHATE to preserve a diffusion-based information distance, where the long-term transition probability catches the global information. Intuitively, PHATE generates a random walk on the data points with the transition probability matrix $P$ where $P[i|j] \propto e^{\gamma \|x_i - x_j\|}$ for some global rescaler $\gamma$.. After $t$ walks, a large transition probability $p^t[i|j]$ where $P^T = \prod P$ will imply that $i$ and $j$ are relatively closer than those pairs of low transition probability. The diffusion-based design of PHATE is especially effective to visualize biological data, which has some development according to time, such as single-cell RNA sequencing of human embryonic stem cells.

**Distance to Measure.**   Distance to measure (DTM) was first introduced by Chazal et al. (2011) and its limiting distributions are investigated in (Chazal et al., 2017). These works are in the context of topological data analysis, where DTM is a robust alternative of *distance to support* (DTS). DTS is a distance from a point $x$ to the support of the data distribution, not a distance between two points. DTS is used to reconstruct the data manifold, for example by $r$-offset (a union of radius $r$-balls centered at the data points), or together with persistent topology (increasing $r$) to infer topological features such as Betti numbers. DTM is introduced as an alternative to DTS because the empirical estimator of DTS depends on only one observation in the dataset, which might be sensitive to outliers. These works do not share the same context with our work

because we are not aiming to infer topological features or recover topology. However, we found that there is a mathematical connection between our local distance rescale and DTM. By connecting the two, we can easily obtain the consistency of our geodesic distance estimators by their limiting distribution theorem.

**Topological Autoencoder** Moor et al. (2020) propose Topological Autoencoder as a new way of regularizing the latent representations of autoencoders by forcing that the topological features (persistence homology) of the input data be preserved in the latent space, adding the interpretability of autoencoders in terms of meaningful latent representation. The information of persistence homology is captured by calculating distances between persistence pairings, so that such distances of the input space and latent space are regularized to be similar. Besides the motivation of autoencoder regularization, Moor et al. (2020) show competing two-dimensional latent space visualizations against existing manifold learning (visualization) methods. Therefore, we compare our iGLoMAP with TopoAE.

**Deep isometric manifold learning** Pai et al. (2019) proposes *Deep isometric manifold learning* (DIMAL), which is an extension of Isomap to exploit deep neural networks to learn a dimensional reduction mapper. DIMAL shares the same spirit of MDS as Isomap does in the sense that it seeks to minimize the quadratic error between the geodesic distance in the input space and the $L_2$ distance on the lower dimensional representation. DIMAL uses a network design called Siamese networks (or a twin network). As the geodesic distance to be preserved in DIMAL is the same as that in Isomap, the key difference between GLoMAP and DIMAL is the key difference between iGLoMAP and Isomap; the geodetic distance estimates and the loss function (the quadratic loss and KL-divergence loss).

**Parametric UMAP** Sainburg et al. (2021) extend UMAP to a parametric version, which we call PUMAP, using a deep neural network such as a autoencoder. PUMAP shares the same graphical representation of a dataset with UMAP, but the optimization process is modified to exploit a mini-batch based stochastic gradient algorithm for training a deep neural network. Like Sainburg et al. (2021), we also use a deep neural network to learn a dimensional reduction function. The key difference between GLoMAP and UMAP is also the key difference between iGLoMAP and PUMAP.

**EVNet** Zang et al. (2022) propose EVNet, a dimensionality reduction method that utilizes deep neural networks, with the purpose of 1) preserving both global and local data information, and 2) enhancing the interpretability of the dimensionality reduction results. EVNet maps the input data twice: first, to a latent space, and then from this space to a lower-dimensional (visualization) space. The distances in the visualization space are regularized to be similar to those in the latent space. We believe that comparing our method with EVNet would be interesting, pending the availability of its public implementation.

**Predictive principal component analysis** Isomura & Toyoizumi (2021) propose *predictive principal component analysis*, which finds a lower-dimensional representation at time $t+1$ as a function of the sequence of dependent data points of time $\{1, ..., t\}$. The lower dimensional representation then is used for predicting the data point at time $t + 1$. It is an interesting problem to condense the dependency structure among data points as a time series into a single lower dimensional representation for the purpose of prediction. We think extending our framework to handle this problem would be an interesting direction.

**VAE-SNE** Graving & Couzin (2020) propose VAE-SNE that incorporates t-SNE to regularize the latent space of the variational autoencoder (VAE). VAE-SNE is similar to Moor et al. (2020) in that it seeks to regularize the latent space of an autoencoder to have a meaningful and probably more interpretable latent representation. The regularization term of VAE-SNE that is added to the (variant of) VAE loss is a probabilistic expression of the t-SNE loss. Therefore, the design of VAE-SNE aims to learn an autoencoder that has a latent representation similar to t-SNE. In our numerical study, we focus on comparing with t-SNE.

## S3   The Datasets

We give details about the datasets used in our numerical study. The Python package of iGLoMAP includes the data generating code for all simulated datasets.

**Scurve**   We adopted the Scurve dataset from the Python package scikit-learn (Pedregosa et al., 2011). Let $t$ be a vector of uniform samples from $(-\frac{3\pi}{2}, \frac{3\pi}{2})$. Also, let $u$ be uniform samples from $(0, 2)$ of the same size. We can think that the original data in the lower dimensional space (2-dimensional) is a matrix $[t, y]$. Now we transform this data matrix into higher dimension (3-dimensional). The first dimension $x$ is defined by $x = \sin t$, where the sin is an element-wise sin operator. The second dimension is simply $y = u$. The third dimension is defined by a vector $z = \sin(t) * (\cos(t) - 1)$. The resulting 3-dimensional data is a matrix $[x, y, z]$. We use $n = 6000$.

**Severed Sphere**   The Severed Sphere dataset is also obtained by the Python package scikit-learn (Pedregosa et al., 2011). Let $u$ be a vector of uniform samples from $(-0.55, 2\pi - 0.55)$. Also, let $t$ be a vector of uniform samples from $(0, 2\pi)$. The 2-dimensional data matrix $[t, y]$ is transformed into three dimension first by transform, and then by point selection. First, the transformation is $[x, y, z] = [\sin(t)\cos(u), \sin(t)\sin(u), \cos(u)]$. Now only the rows of the data matrix such that $\frac{\pi}{8} < t < \frac{7\pi}{8}$ are selected for $t \in t$. We use $n = 6000$.

**Eggs**   The Eggs data is made up of 5982 data points in the three-dimensional space. It has 12 half-spheres (empty inside) and a rectangle with 12 holes. When we locate the 12 half-spheres on the holes on the rectangle, we obtain the shape of a two-dimensional surface, where the half spheres and the boundaries of the wholes are seamlessly connected. On the rectangular surface, the points on the flat area are little sparser than in the curved area; we uniformly distribute 2130 points on the rectangle $[-16, 16] \times [-4, 4]$ and remove the points on the 12 circles with radius 1, centered at $[(-13, -2), (-13, 2), (-8, -2), (-3, 2), (2, +2), (7, -2), (12, 2), (-8, +2), (-3, -2), (2, -2), (7, +2), (12, -2)]$. Depending on the random uniform generation, the amount of points removed can vary. Therefore, in our setting, we use a fixed random generating seed number so that the training set is always the same, and always the number of remaining points is 1266. Each half-sphere of radius 1 has 348 uniformly distributed points. Therefore, the dataset in total has $12 \times 348 + 1266 = 5982$.

**Hierarchy**   The hierarchy dataset is developed by Wang et al. (2021). The five macro centers are generated from $N(0, 100^2 I_{50})$ where $I_{50}$ is a 50x50 identity matrix. We call each macro center as $M_i$ for $i = 1, ..., 5$. Now, for each macro center, 5 meso centers are generated from $N(M_i, 1000 I_{50})$. We call the meso centers as $M_{ij}$ where $j = 1, .., 5$. Now, for each meso center, 5 micro centers are generated from $N(M_{ij}, 100 I_{50})$. We call the micro centers $M_{ijk}$, where $k = 1, ..., 5$. Now for each micro center, a local cluster is generated from $N(M_{ijk}, 10 I_5)$. In our experiment, each local cluster consists of 48 points so that in total $n = 6000$.

**Spheres**   The spheres dataset is designed by Moor et al. (2020). It has one large outer sphere on which half of the dataset is uniformly distributed and ten smaller inner spheres on which the remaining points are uniformly distributed. Let the center an inner sphere be denoted by $u_i$, where $i = 1, ..., 10$. Then, $u_i$ is generated from $N(0, 0.5 I_{101})$. Now, 5% of the dataset is uniformly distributed on the sphere of radius one centered at $u_i$. The large outer sphere is centered at the origin and has a radius of 25. The 50% of the data set is uniformly distributed on this outer sphere. In our case, we use $n = 10000$.

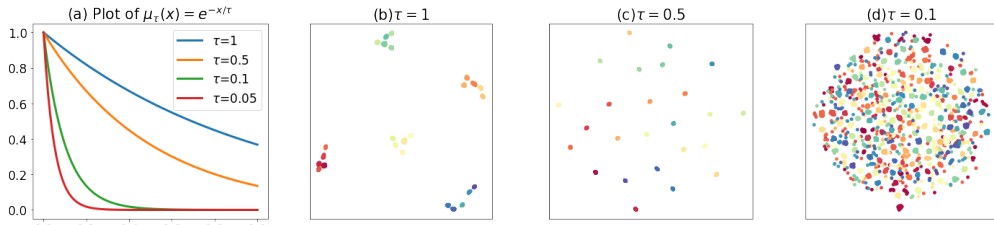

Figure S10: GLoMAP *without* $\tau$ tempering on the hierarchical dataset. A larger *fixed* $\tau$ finds more global shapes, and smaller *fixed* $\tau$'s find more local shapes while missing global clusters.

## S4   Learning Configurations

**Learning Configuration for Example 1.**   We apply UMAP and GLoMAP, where for both methods, the number of neighbors is set as $K = 15$ (default) for the spheres, and as $K = 250$ for the hierarchical dataset. The latter is large to make the graph connected at least within each of the macro-level clusters. We fix all $\lambda_e$ to 1 (default), while $\tau$ is scheduled to decrease from 1 to 0.1 (default).

**Learning Configuration for Section 5.**   Typically, we set $\lambda_e$ at 1 (default) to showcase performance without tuning. However, for MNIST, we opt for tighter clustering by setting $\lambda_e$ at 0.1. In the case of iGLoMAP applied to the S-curve, the shape initially remains linear, then expands suddenly at the end. To address this, we adjusted the starting $\tau$ to be four times smaller as outlined in Remark 1. For similar reasons, we made adjustments on the Eggs and Severed datasets, allowing shapes to form earlier and providing sufficient time for detail development. In the MNIST dataset, where both GLoMAP and iGLoMAP tend to linger in the noise, we reduced the starting $\tau$ by a quarter. Apart from these specific cases, we maintain the default $\tau$ schedule (from 1 to 0.1). All other learning hyperparameters are set to their defaults in the iGLoMAP package (learning rate decay = 0.98, Adam's initial learning rate = 0.01, initial particle learning rate = 1, number of neighbors K=15, and mini-batch size=100). GLoMAP was optimized for 300 epochs (500 epochs for MNIST), and iGLoMAP was trained for 150 epochs.

## S5   Additional Discussions

### S5.1   The effect of tempering

In Example 1, it seems tempering is vital to discover the hierarchical clustering structure in the data. Figure S10 (a) depicts the effect of decreasing temperatures on $\mu^\tau(d) = \exp\{-d/\tau\}$. For a given $d$, a decreasing $\tau$ reduces $\mu^\tau(d)$, requiring the corresponding pair on the visualization space to be more distanced. This makes the local clusters stay close at the early stage while forming the global shape, and separates them at the late stages, forming the detailed local structures. The same progression from global shape to detailed localization does not happen without the tempering (when $\tau$ is kept as a constant) as Figure S10 (b-d). A larger fixed $\tau$ ($\tau = 1$) finds global shapes, and smaller fixed $\tau$'s find more local shapes while missing the global clusters.

### S5.2   The effect of the aesthetics parameter $\lambda_e$

For Example 1, to see the effect of the aesthetics parameter $\lambda_e$ in controlling the attractive and repulsive forces, we vary $\lambda_e$ for a wide range in Figure S11 and S12. We observe that similar development patterns occur across a wide range of $\lambda_e$ values. However, a smaller $\lambda_e$ more strongly condenses data points within the cluster, and vice versa. For an extremely large $\lambda_e$, the cluster shape may not appear, suggesting a possibility that optimization did not occur properly.

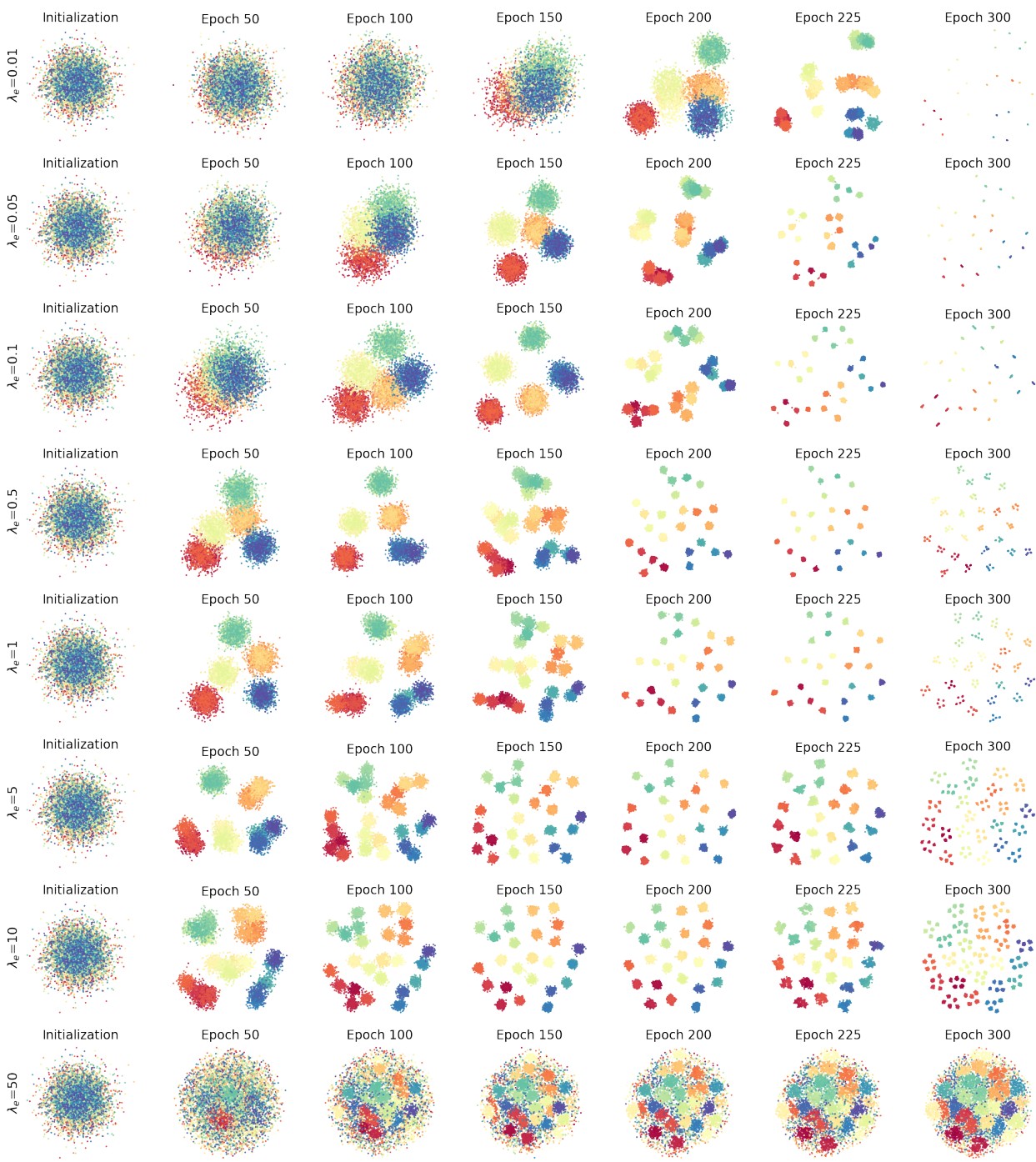

Figure S11: The sensitivity of $\lambda_e$ of the transductive GLoMAP on the hierarchical dataset (Example 1). A larger $\lambda_e$ tends to make a local cluster more spread. However, the general tendency of the development from global structures to local structures is similar across a wide range of $\lambda_e$ values. The $\tau$ scale was reduced from 1 to 0.1

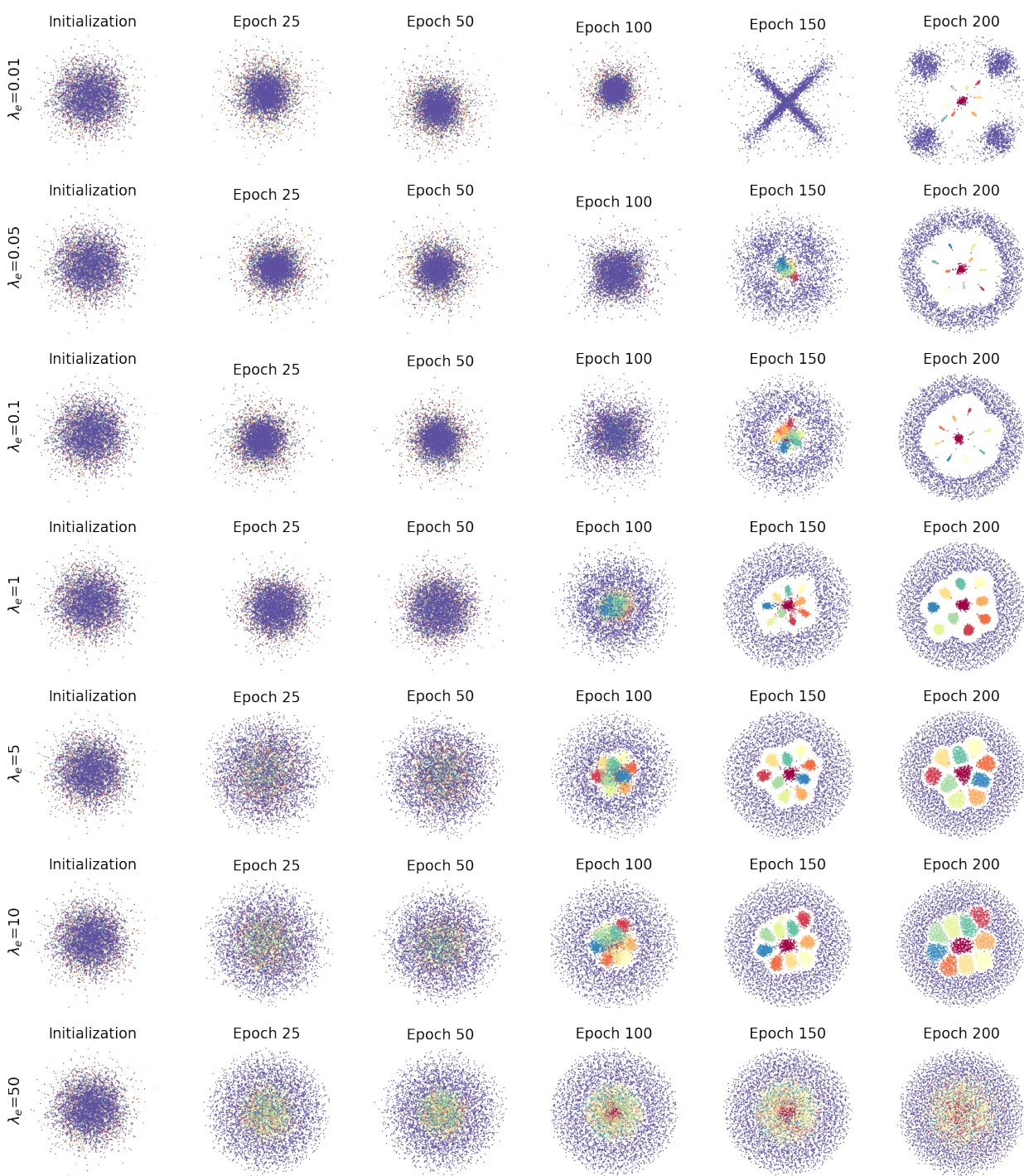

Figure S12: The sensitivity of $\lambda_e$ of the transductive GLoMAP on the spheres dataset (Example 1). A larger $\lambda_e$ tend to make a local cluster more spread. However, the general tendency of the development from global to local structures is similar across a wide range of $\lambda_e$ values. The $\tau$ scale was reduced from 1 to 0.1.

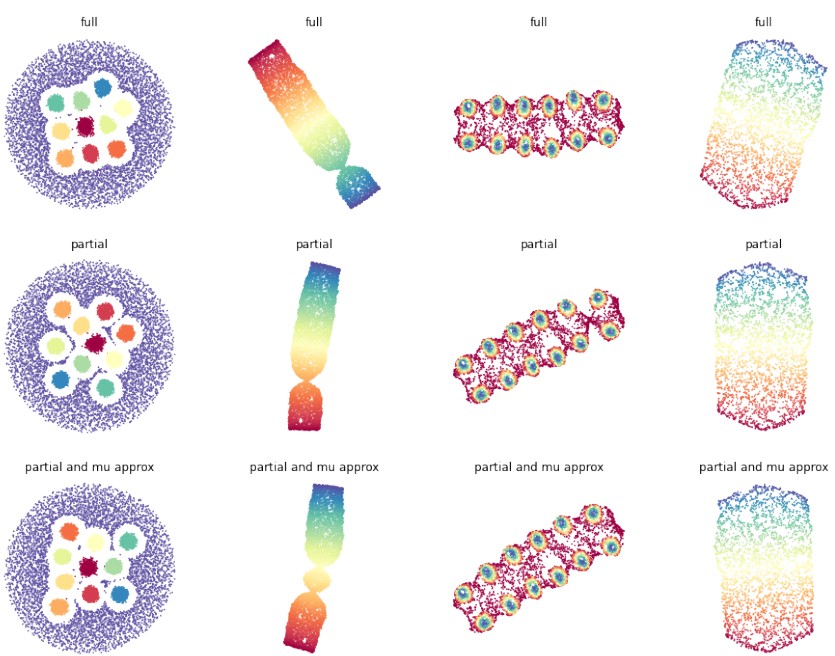

Figure S13: GLoMAP faster variants results (Section S5.3). Top: the exact loss, Middle: simplified $D_{\text{glo}}$ is used ($\tilde{K}$-nearest neighbor distance matrix of $D_{\text{glo}}$ is used instead of $D_{\text{glo}}$), Bottom: simplified $D_{\text{glo}}$ is used and $(1 - \mu_{ij})$ is approximated by 1. Number of data points: Spheres 10,000, S-curve 8,000, Eggs 6,000, Severed 4,000. We set $\tilde{K} = 15m$, where $m = \lfloor n/100 \rfloor$.

### S5.3 Computational cost reduction

While we have reduced the computational cost for local distance estimation, managing the global distances, conversely, increases the cost. Besides the shortest path search, the neighbor sampling scheme (line 9 in Algorithm 2) incurs costs comparable to the normalization of t-SNE; The sampling involves summing membership scores over $n$ distances, given that all data points are connected. The cost of the sampling scheme can be reduced by instead considering the $\tilde{K}$-nearest neighbor distance matrix of $D_{\text{glo}}$, for example, with $\tilde{K} = 20K$. To further reduce the computational cost, we could also ignore the difference between $(1 - \mu_{ij})$ values, but consider them all as 1. This saves time in retrieving the $\mu_{ij}$ values from the $n \times n$ (sparse) matrix. Since $(\mu_{i\cdot})$ is only an $n \times 1$ vector, it does not take much time to look up. In such a case, the loss function is $\hat{\mathcal{L}} = -\sum_{i \in S} \mu_{i\cdot} \log q_{ij_i} - \lambda_e \sum_{i \in S} \sum_{j \in S} \log (1 - q_{ij})$. For examples of these approximations' results, see Figure S13 for the visualization and Figure S14 for the computational time comparison. These figures demonstrate a high similarity to the results obtained from the exact loss computation, and the reduction in computational cost becomes more evident for larger datasets.

## S6 Additional Experiments

In this section, we discuss the performance of our methods on an interesting dataset, called Fishbowl. In addition, we also empirically investigate the sensitivity of iGLoMAP against the capacity choice of deep neural networks for the mapper.

**Fishbowl Data.** It is an interesting problem how a dimensional reduction method performs on a dataset, which is not homeomorphic to $\mathbb{R}^2$, for example, a sphere. In such a case, there is no way to map onto the two dimensional space while preserving all the pairwise distances in the data. Now, we slightly change the problem. We consider that we are given data on a half-sphere. This is the famous crowding problem, and in this case also, there is no way to map onto the two dimensional space while preserving all the pairwise distances. Intuitively, we may want to smash (or, stretch) the half sphere to make it flat on the

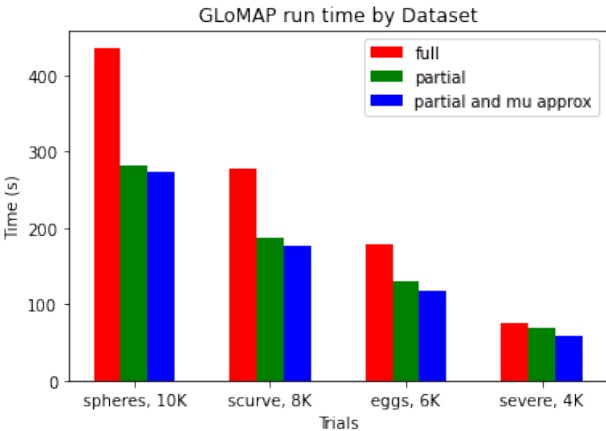

Figure S14: Run time comparison of GLoMAP and faster variants in Section S5.3. Full: the exact loss, Partial: $\tilde{K}$-nearest neighbor distance matrix of $D_{\text{glo}}$ is used instead of $D_{\text{glo}}$, Partial and mu approx: $\tilde{K}$-nearest neighbor distance matrix and $(1 - \mu_{ij})$ is approximated by 1. The effect of using $\tilde{K}$ becomes more evident for larger datasets.

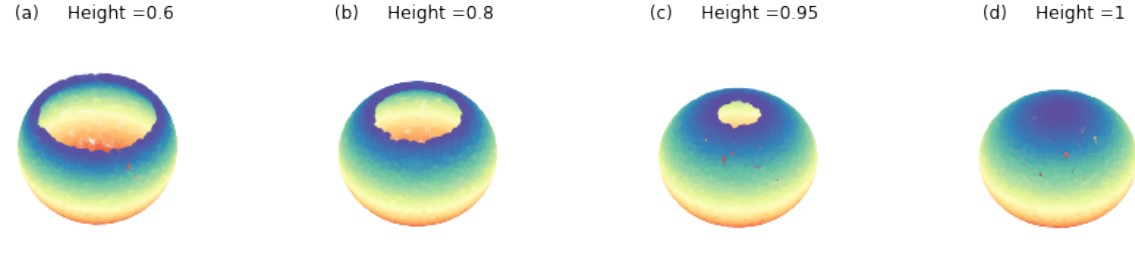

Figure S15: The fishbowl dataset (n=6000).

two dimensional space. Now, we turn our attention to a dataset on something between a sphere and a half-sphere, called a fishbowl (Silva & Tenenbaum, 2002), which has more than half of the sphere preserved. We generate a uniformly distributed dataset on a fishbowl $F_\gamma = \{(x, y, z) \in \mathbb{R}^3 | x^2 + y^2 + z^2 = 1, z \leq \gamma\}$ for $\gamma \in \{0.5, 0.6, 0.7, 0.8, 0.9, 0.95, 0.975, 1\}$. The data is illustrated in Figure S15. When $\gamma = 1$, the data are equivalent to a uniform sample on a sphere. Although there is no consensus on what type of visualization must be ideal, in this case the exact pairwise distance is impossible. All learning parameters and hyperparameters are set to their default values, except for the initial $\tau$ value for iGLoMAP, which was reduced to 0.5. This adjustment was necessary because, under the default value, iGLoMAP remains in a linear (non-meaningful) state until very late (see Remark 1). The results of GLoMAP and iGLoMAP are shown in Figure S25. For GLoMAP, we see some recovery of the original disk until $\gamma \leq 0.95$. Then, when $\gamma \geq 0.975$ (when the fishbowl is closer to a sphere), GLoMAP simply shows a side image. For iGLoMAP, the flat disc shape is made until $\gamma = 0.975$. The result on a sphere is in Figure S25 (e) for GLoMAP and Figure S25 (j) for iGLoMAP.

**Sensitivity analysis on the deep neural network capacity** In order to see the impact of network size, we use the same fully connected deep neural network (four layers) with varying network width. The network width (which controls the network size) varies in $\{2^q | q = 2, 4, 6, 8, 10, 11\}$, of which the range is intentionally chosen to be extremely wide. We enumerate the corresponding capacities as 0 to 5. Width 4 is impractically small; When networks are this small, the network may not be expressive. Width 2048 is the extreme opposite. Note that width 4 has 66 parameters to learn in the network, width 16 has 646, width 64

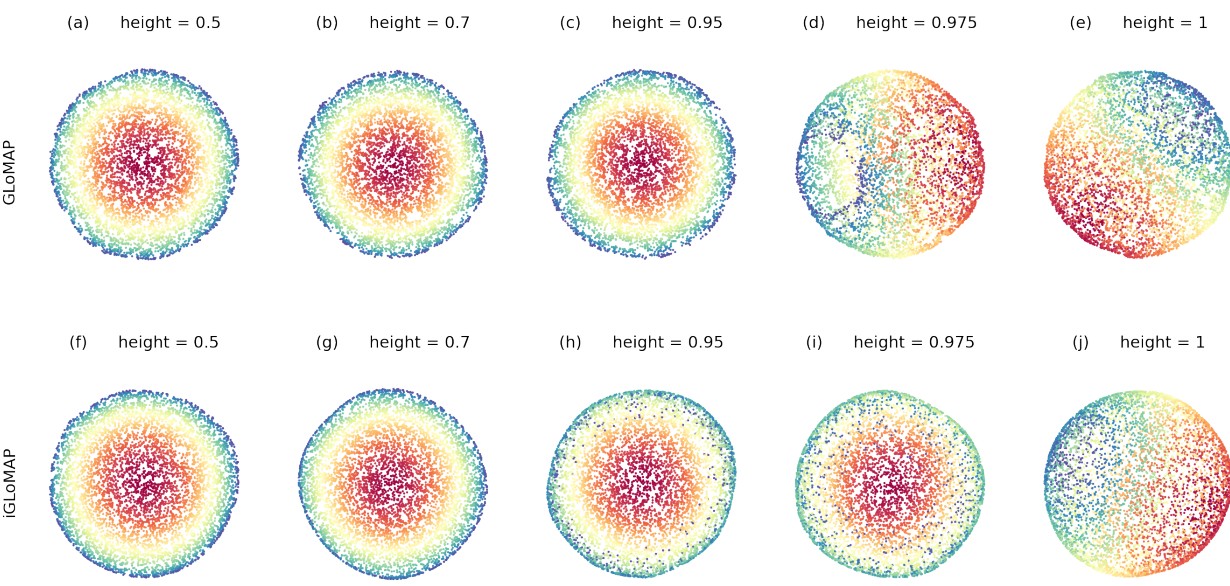

Figure S16: The result of GLoMAP on the fishbowl dataset. The number on each panel indicates the height of fish bowl (the threshold on the z-axis).

has 8,706, width 256 has 133,122, width 1024 has 2,105,346, and width 2048 has 8,404,994. We recall that in Section 5, we used width 64 for all simulation datasets. We set the hyperparameters of iGLoMAP same as in Section 5.2, except halving the starting $\tau$ to 0.5 for Spheres since for smaller networks, the expansion of visualization is observed too late.

The visualizations of the results are in Figure S17. We see that increasing the network size improves the visualization quality from width 4 to width 64. On the other hand, among larger networks, we see very similar results, concluding that for a good range of network widths the visualizations are fairly similar. Therefore, we see that for a wide range of network width (of the fully connected deep neural network that we use here), the performance of iGLoMAP is stable. Those visual inspections are supported by the numerical measures shown in Figures S18 and S19, which generally show slightly better performance of a larger network. This result implies that we should avoid using too small networks, but much less caution is needed about using too large networks.

| Capacity 0 | Capacity 1 | Capacity 2 | Capacity 3 | Capacity 4 | Capacity 5 |
|---|---|---|---|---|---|

Figure S17: Sensitivity analysis on the deep neural network capacity (n=6000). The same fully connected deep neural network (four layers) is used with varying layer width. The range of width is intentionally extreme (from 4 to 2048).

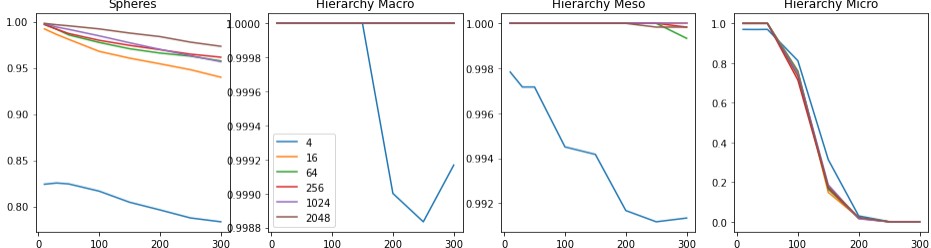

Figure S18: Sensitivity analysis on the deep neural network capacity. x-axis: the number of neighbors $K$. y-axis: the KNN measure as analogous to Figure 7. The numbers in the legend are the layer width of the corresponding deep neural network. From Spheres, we see that a larger capacity is generally better.

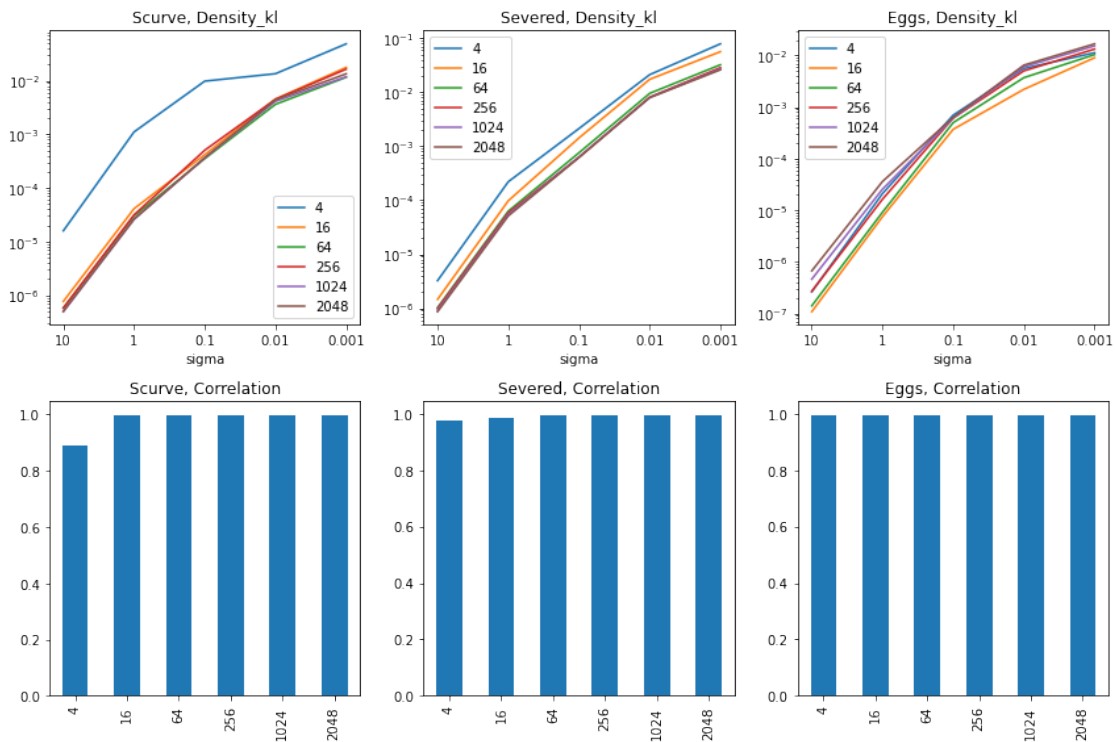

Figure S19: Sensitivity analysis on the deep neural network capacity. The KL divergence (first row) and correlation (second row) defined in Section 5.1.1. The numbers in the legend are the layer width of the corresponding deep neural network. From S-curve and Severed Sphere, it looks a larger capacity is generally better, while the performance measures show similar patterns.

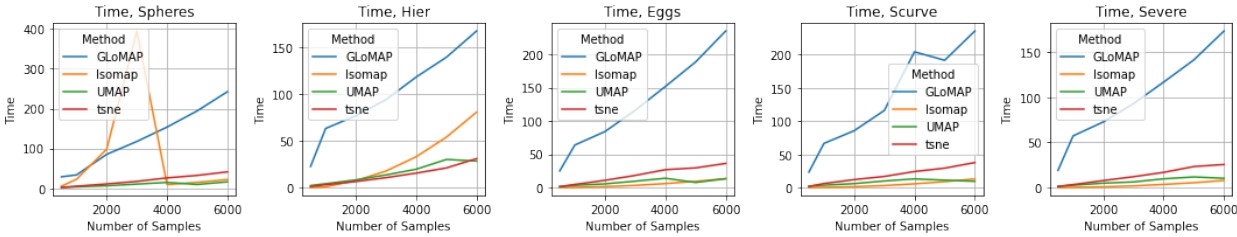

Figure S20: Comparison of computational time. There is room for improvement in the GLoMAP implementation, as it currently relies solely on Python without optimization through languages/interfaces like Numba, Cython, or C++. Additionally, the neighborhood search algorithm is a basic version rather than a faster approximation.

## S7    Additional Measures

In Van Assel et al. (2024), the Silhouette coefficient (Rousseeuw, 1987) and the trustworthiness score (Venna & Kaski, 2001) are used to assess the quality of dimensional reduction. The Silhouette coefficient measures both the cohesion within clusters and separation between clusters, defined as $b - a/\max(a, b)$, where $a$ is the mean intra-cluster distance, and $b$ is the mean nearest-cluster distance. The trustworthiness score quantifies how well the K-nearest neighbors in the low-dimensional representation aligns with the rank of among the Euclidean distances in the input space. Both scores are computed using the Scikit-learn Python package Buitinck et al. (2013).

The full results are shown in Figure S27 and S26, while Table S1 shows the case when $n = 6000$ with $K = 5$ for trustworthiness measure. In Figure S27, For the Hierarchical dataset, when $K$ is small, Isomap performs poorly under the Trustworthiness measure, while for larger $K$ values, UMAP exhibits lower performance. GLoMAP demonstrates competitive performance overall. This consistent competence is also shown in the Silhouette coefficient as in Figure S26. For the Spheres dataset, however, GLoMAP performs the worst on this Trustworthiness measure and improves on the Silhouette coefficient. The corresponding visualization is presented in Figure S21, which we believe remains acceptable or even desirable.

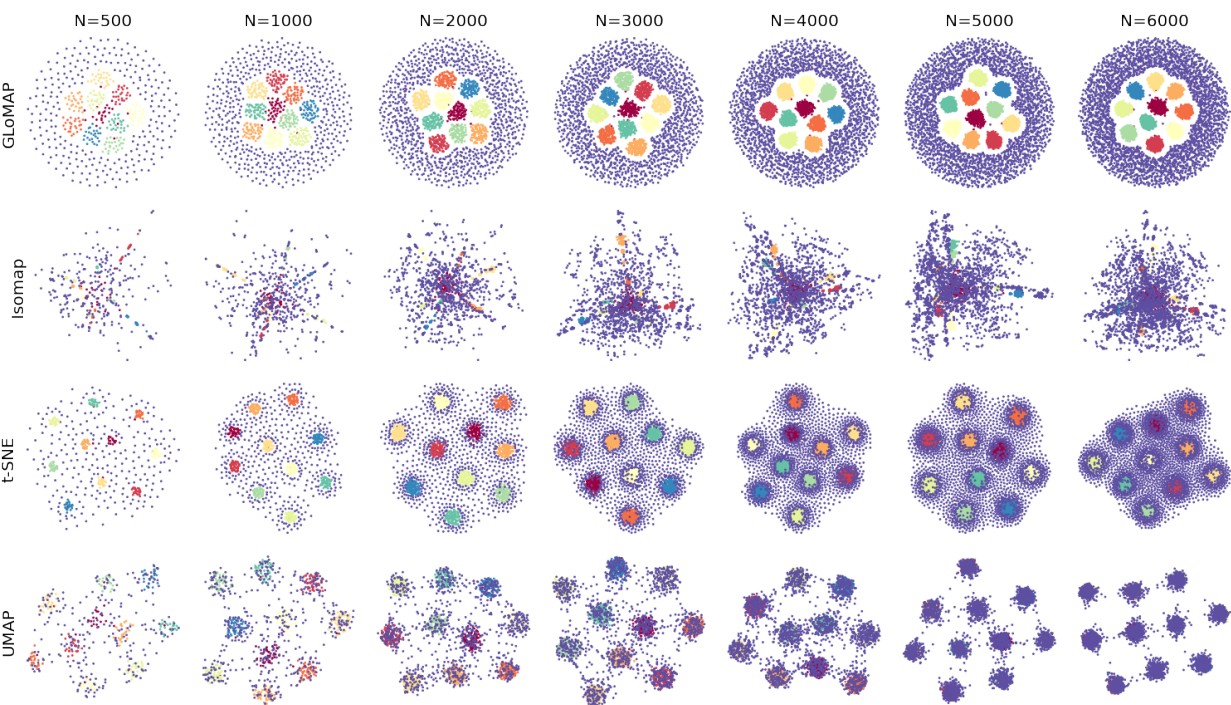

Figure S21: Comparison of visualizations for increasing data sizes on the Spheres dataset. The corresponding Silhouette coefficients and trustworthiness scores are presented in Figure S27 and Table S1.

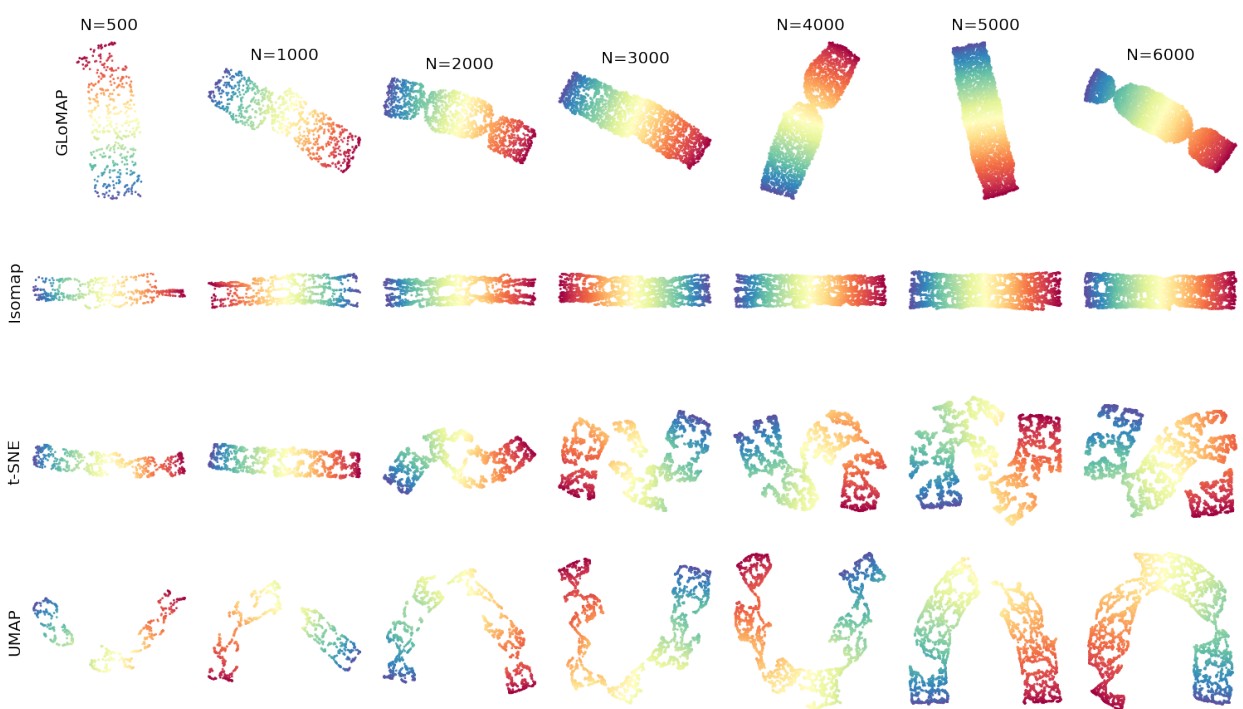

Figure S22: Comparison of visualizations for increasing data sizes on the Scurve dataset. The corresponding Silhouette coefficients and trustworthiness scores are presented in Figure S27 and Table S1.

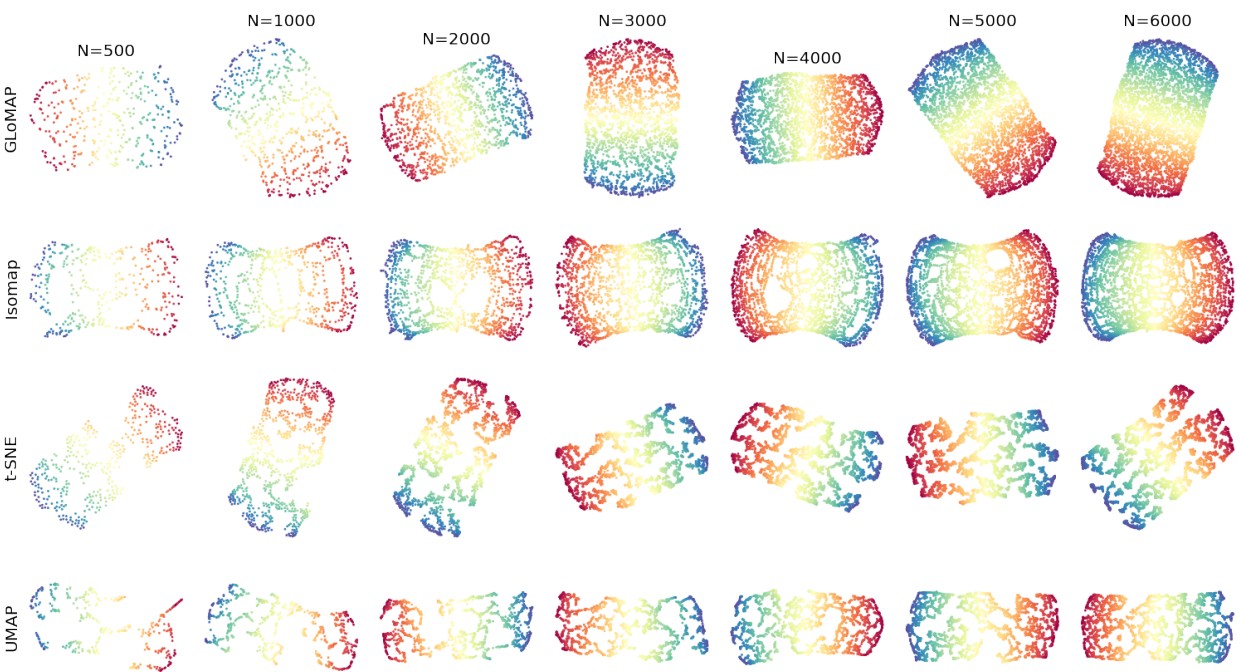

Figure S23: Comparison of visualizations for increasing data sizes on the Severed dataset. The corresponding Silhouette coefficients and trustworthiness scores are presented in Figure S27 and Table S1.

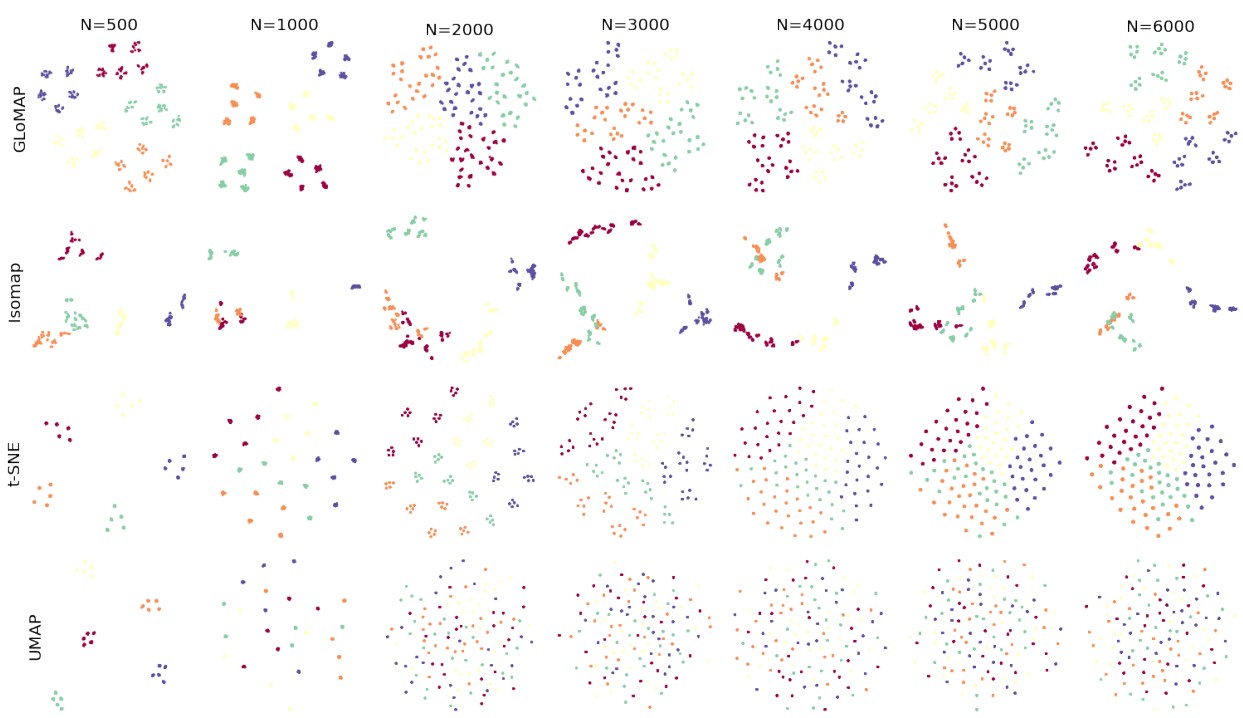

Figure S24: Comparison of visualizations for increasing data sizes on the Hierarchical dataset. The corresponding Silhouette coefficients and trustworthiness scores are presented in Figure S27 and Table S1. For GLoMAP, Isomap, UMAP, to make the graph connected among higher-level clusters, we use n_neighbors=250.

| | Trustworthiness | | | | Silhouette | | | |
|---|---|---|---|---|---|---|---|---|
| Method | GLoMAP | Isomap | UMAP | t-SNE | GLoMAP | Isomap | UMAP | t-SNE |
| Eggs | 0.998 | 0.999 | 1.000 | 1.000 | . | . | . | . |
| Hierarchical (Macro) | 0.997 | 0.989 | 0.997 | 0.999 | 0.413 | 0.424 | -0.059 | 0.226 |
| Hierarchical (Meso) | 0.997 | 0.989 | 0.997 | 0.999 | 0.741 | 0.614 | -0.198 | 0.059 |
| Hierarchical (Micro) | 0.997 | 0.989 | 0.997 | 0.999 | 0.907 | 0.489 | 0.991 | 0.930 |
| Scurve | 0.998 | 1.000 | 1.000 | 1.000 | . | . | . | . |
| Severed | 0.999 | 1.000 | 1.000 | 1.000 | . | . | . | . |
| Spheres | 0.615 | 0.633 | 0.660 | 0.681 | 0.016 | -0.028 | -0.022 | 0.002 |

Table S1: The trustworthiness score and Silhouette coefficient for various datasets with $n = 6000$. The full results are in Figure S27. The Silhouette coefficient is computed only for the Hierarchical and Spheres datasets, as it requires the class information.

## S8    Additional Visualizations

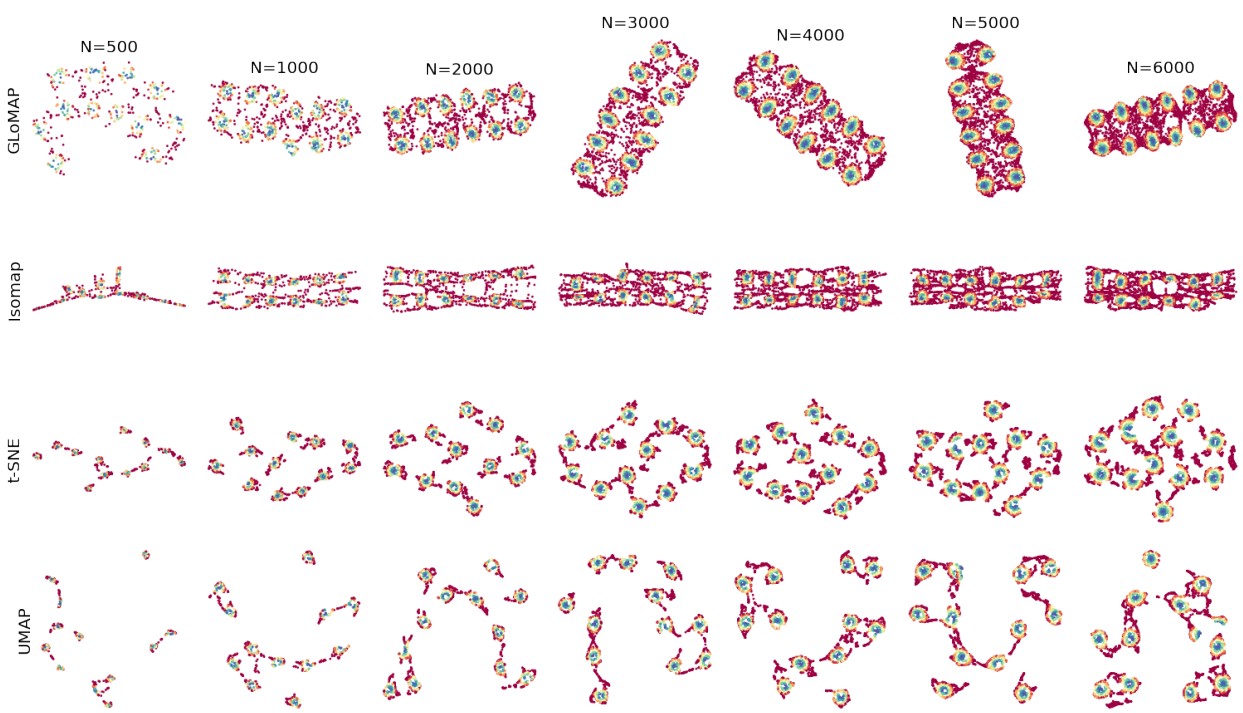

Figure S25: Comparison of visualizations for increasing data sizes on the Eggs dataset. The corresponding Silhouette coefficients and trustworthiness scores are presented in Figure S27 and Table S1.

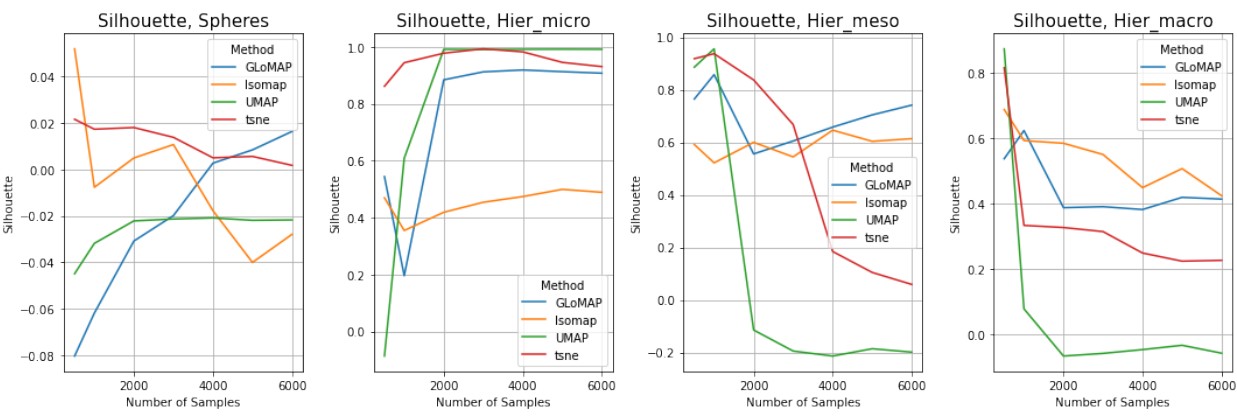

Figure S26: The Silhouette coefficient (Rousseeuw, 1987; Buitinck et al., 2013) are evaluated for an increasing number of samples. For the Hierarchical dataset, GLoMAP shows a consistently competitive result for all levels of clusters.

## S9    Comparison with C-Isomap

To see the usefulness of our distance in equation 6, we plug-in our global distance into the Isomap framework, i.e., apply an MDS onto our global distance matrix. As a baseline, we choose C-Isomap (Silva & Tenenbaum, 2002) among many Isomap variants to improve the local distance because C-Isomap is a scalable option as it also provides a closed-form local distance estimator. C-Isomap applies shortest path search over the local distances and then Multidimensional Scaling (MDS). Since MDS does not allow any pair having as the distance, we imputed 1.5 times of the maximum of pairwise distances for such pairs. As shown in Figure S38, for C-Isomap, when K is large then we see separation between the purple and the other classes. When K is small, the smaller classes are doted, and each class may compose multiple clusters in visualization depending on the size of . On the other hand, when C-Isomap is using our distance, we can see separation.

Our GLoMAP corresponds to Algorithm 1 + Algorithm 2. Our choice of designing the loss similar to that of UMAP (as in Algorithm 2) was also significant, and novel if we see it from the perspective of Isomap. When comparing to Figure 1, we can see that using Algorithm 2 instead of MDS brings a better clustering effect for those ten inner clusters.

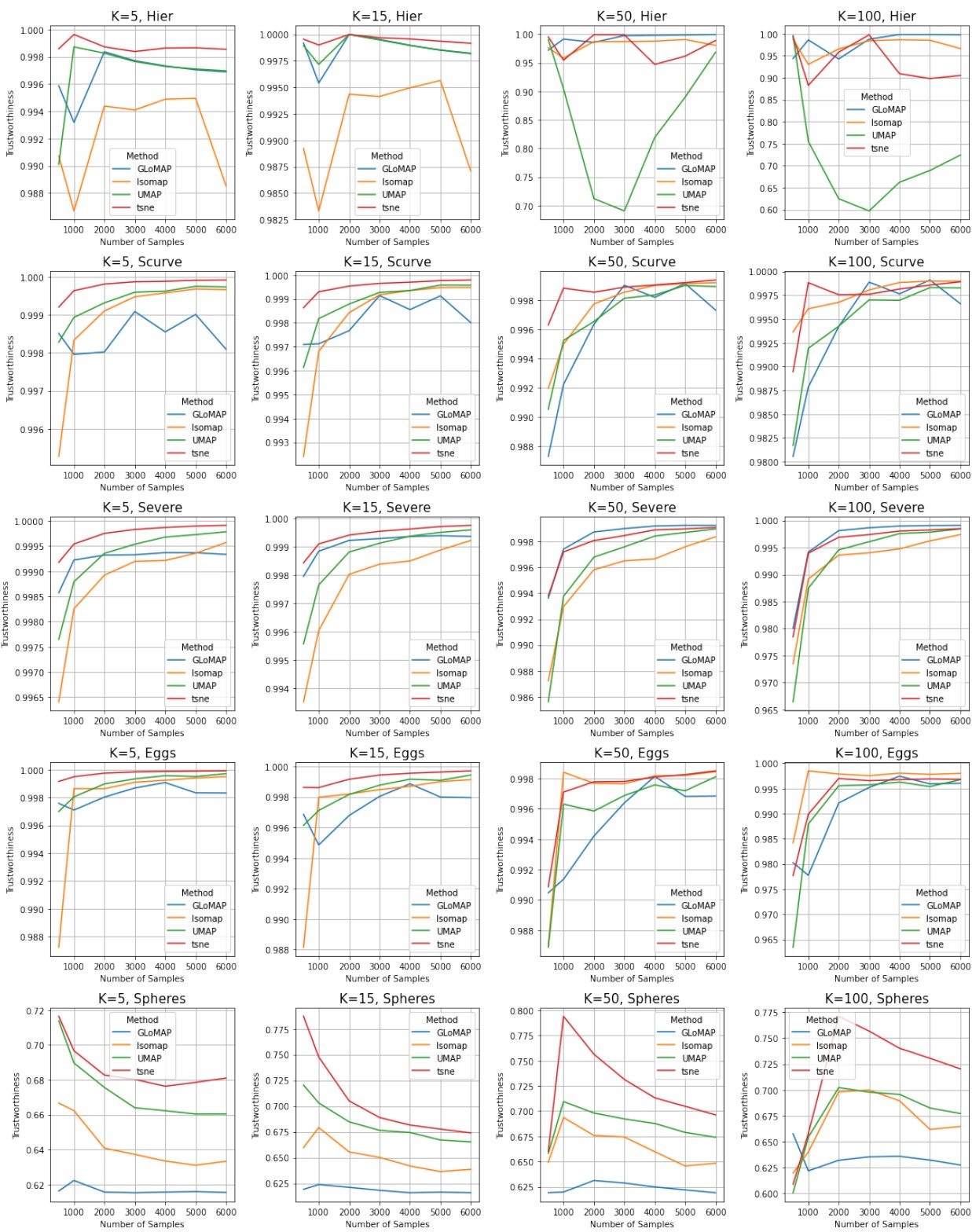

Figure S27: The trustworthiness scores (Venna & Kaski, 2001; Buitinck et al., 2013) are evaluated for an increasing number of samples. This score depends on a hyperparameter, $K$, which measures the correspondence of $K$-nearest neighbors between the input and visualization spaces.

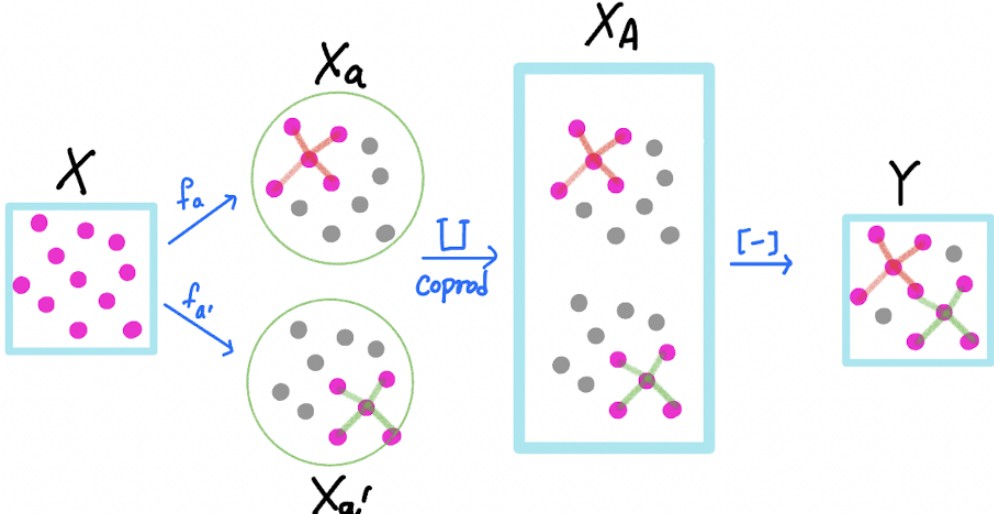

Figure S28: The effect of equivalence relation as a coequalize. For simplicity and visual clarity, we illustrate the coproduct between only $X_a$ and $X_{a'}$, rather than all $n$ pseudo-metric spaces..

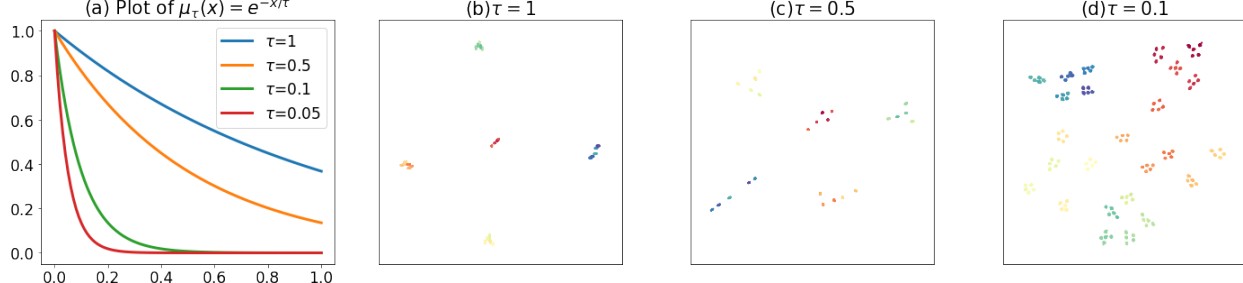

Figure S29: iGLoMAP without $\tau$ tempering. Even when small $\tau$ is used for the entire optimization, the global information is still found.

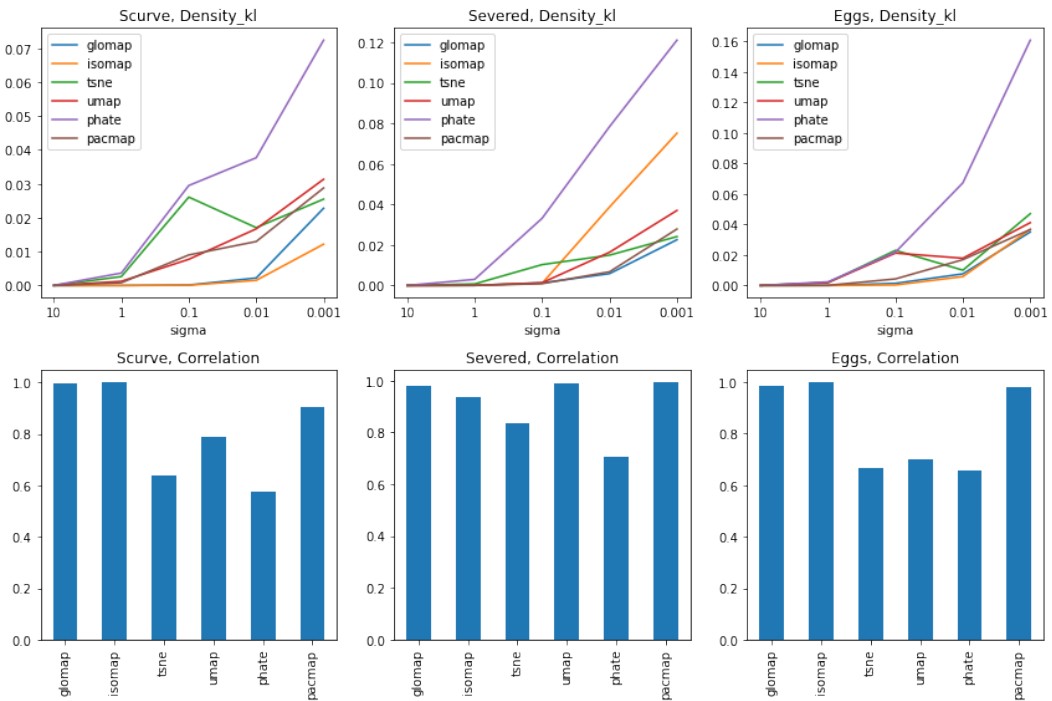

Figure S30: The global preservation measures. Top: $KL_\sigma$ (smaller is better), Bottom: distance correlation (larger is better)

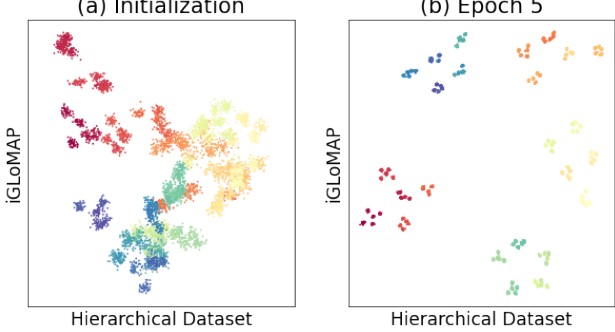

Figure S31: iGLoMAP on the hierarchical dataset using the default $K = 15$ neighbors, which is generally too few for meso level cluster connectivity. However, the macro and meso clusters are still preserved from the start. This suggests the deep neural network might inherently encode some location information. (a) The *random* initialization of the deep neural network. It shows many levels of clusters. (b) At the epoch 5, already all the levels of clusters are found.

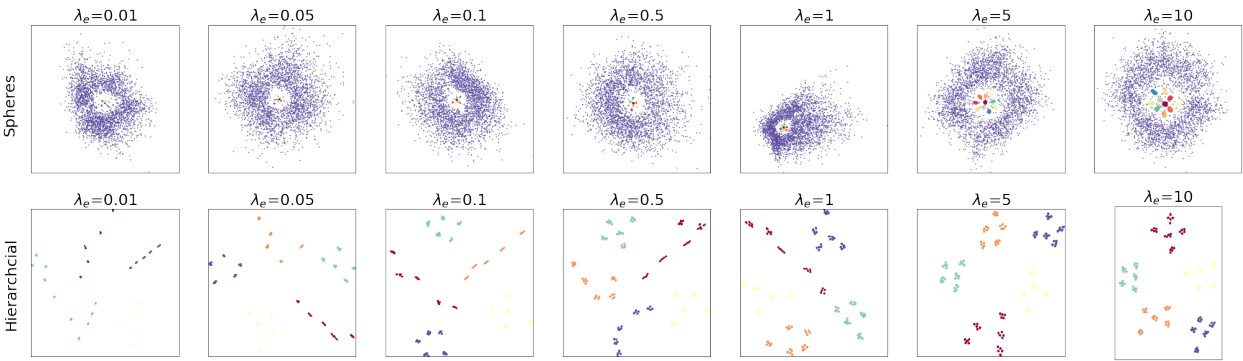

Figure S32: Hyperparameter sensitivity of iGLoMAP. Initial $\tau$ is 1 and sent to 0.1 for all cases (default).

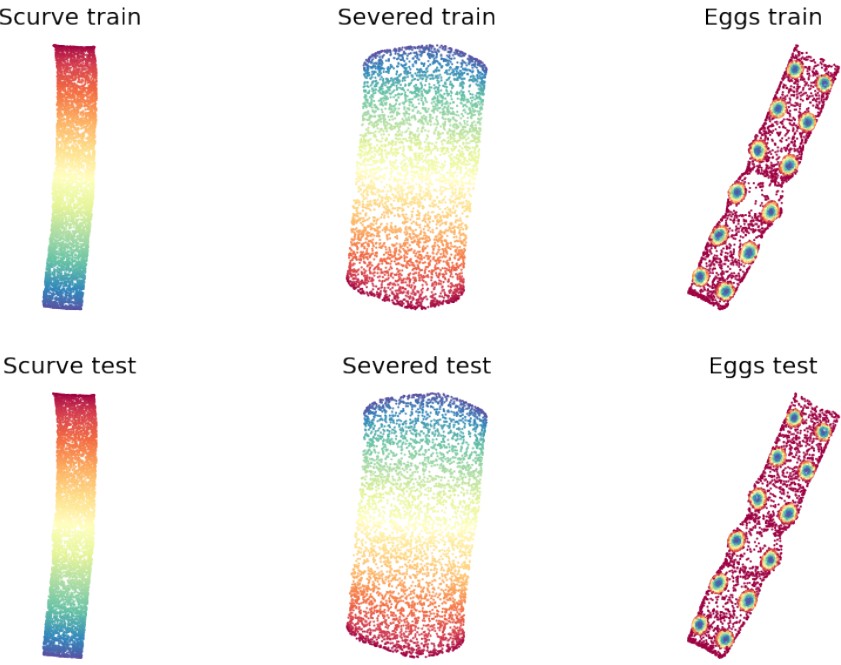

Figure S33: The iGLoMAP generalizations for the S-curve, Severed Sphere, Eggs dataset. The generalization on the test set is almost identical to the original DR on the training set.

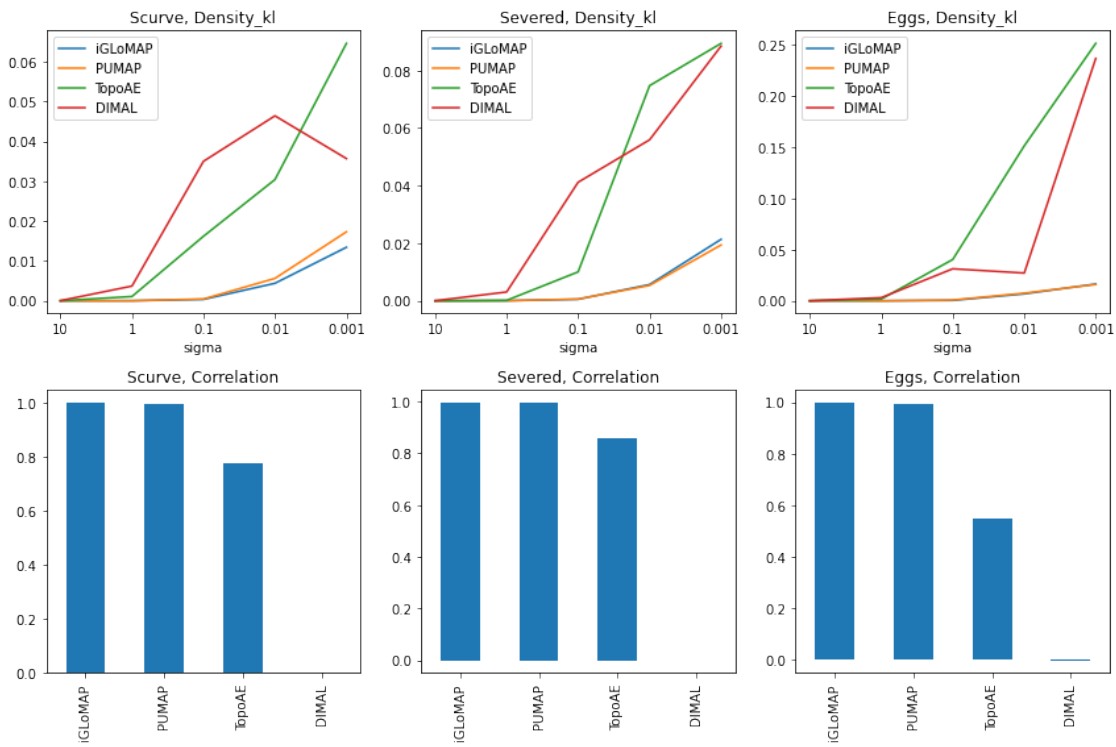

Figure S34: Numerical comparison with other models that utilize deep neural network.

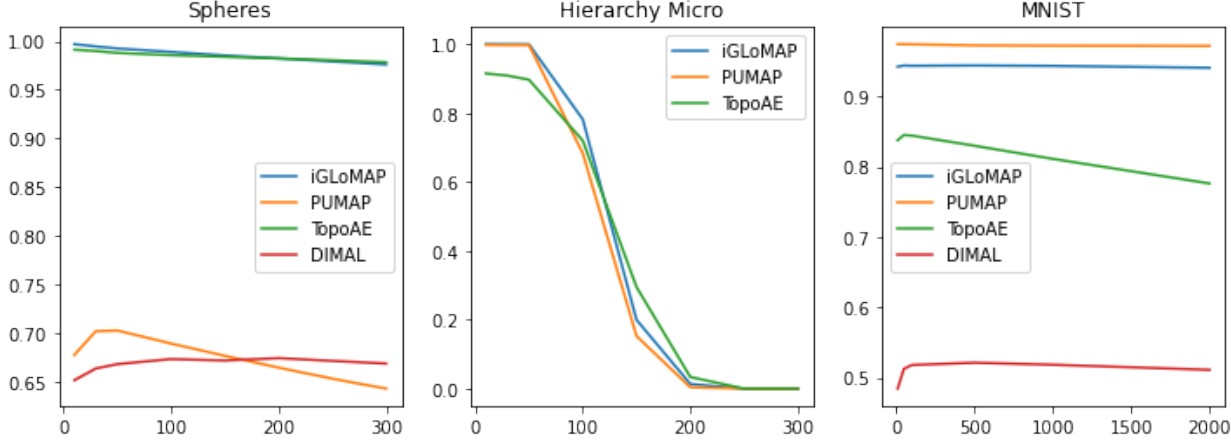

Figure S35: Numerical comparison with other models utilizing deep neural networks. For the hierarchical dataset, macro and meso-level performance metrics are omitted, as iGLoMAP, PUMAP, and TopoAE all achieve 100% performance.

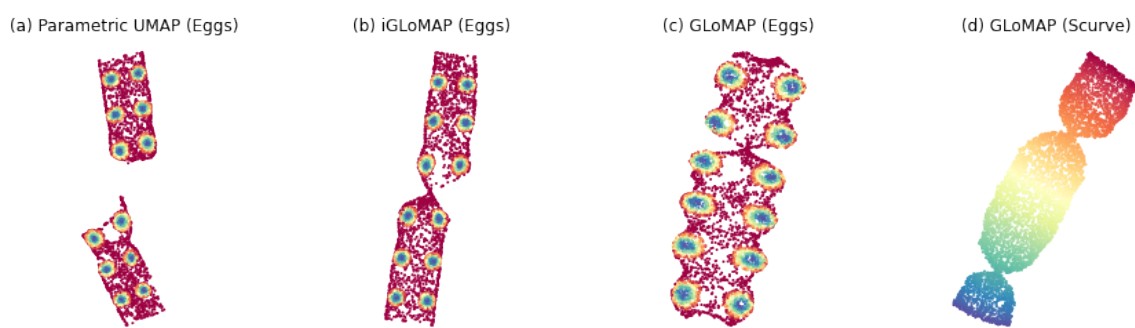

Figure S36: (a) Parametric UMAP: The majority runs of parametric UMAP resulted such a shape, so we cherry picked the best one for comparison as in Figure 8. (b-d) Depending on the run or choice of the hyperparameters, GLoMAP and iGLoMAP may result in a twisted visualization. However, we did not observe any case having two separated partial visualizations.

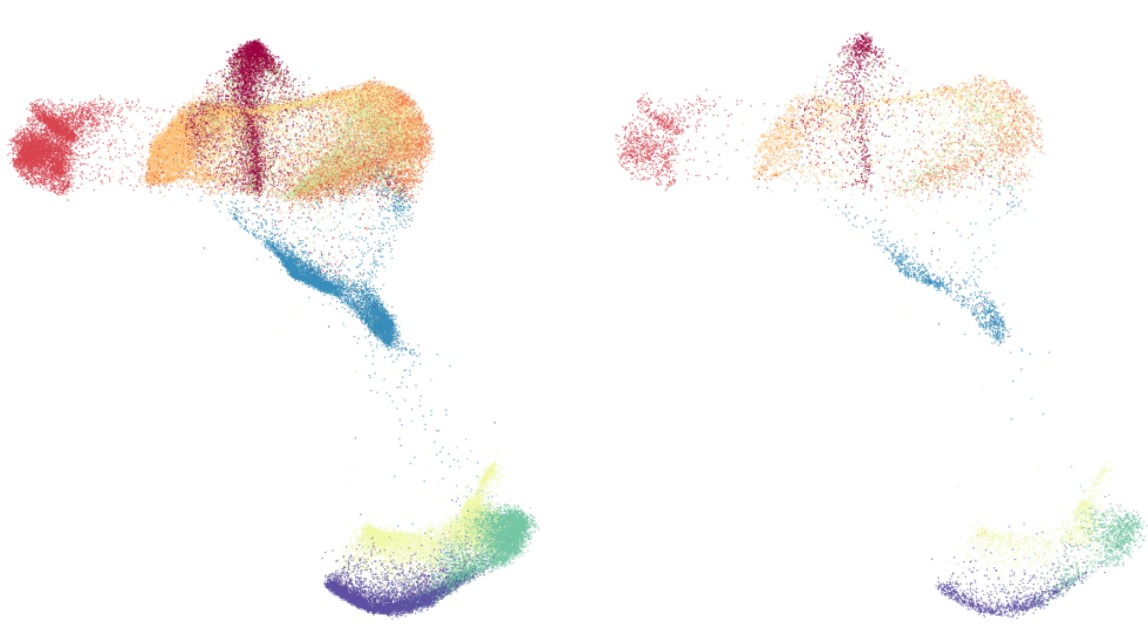

Figure S37: The iGLoMAP result on the Fashion MNIST dataset (Xiao et al., 2017). Left: The visualization result on the training data (n=60000). Right: Generalization result on the test data (n=10000).

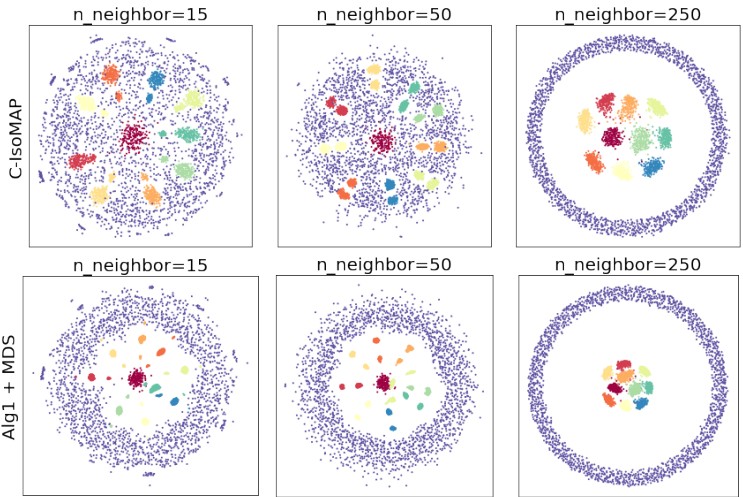

Figure S38: The effect of applying our new global distance to C-Isomap instead of its original global distance estimates, which corresponds to the distance computation by Algorithm 1 and applying MDS in the resulting distance matrix. C-Isomap shows the separation of the purple class when $K$ (n_neighbor) is set as large.

