# OpenReview forum: "Inductive Global and Local Manifold Approximation and Projection"
_TMLR — Accepted by TMLR_

### Review · Reviewer_yBN5 · 2024-10-05

**Summary Of Contributions:**

The submission describes a novel manifold learning method in two versions. The first is a transductive algorithm called GLoMAP and the second an inductive version called iGLoMAP. GLoMAP is a dimensionality reduction technique able to capture both global and local structures. The submission also equips iGLoMAP with a particle-based algorithm to allow for it to be trained in conjunction with a (deep) encoder trained via gradient decent, without the need for additional evaluations/gradient computations of the encoder. It also present theoretical results showing consistency of the new local geodesic estimator, a key part of both algorithms. Finally, it demonstrates the usefulness of GLoMAP and iGLoMAP by comparing its performance against state-of-the-art methods on synthetic and real-world data. The authors present extensive numerical results.

**Audience:**

Yes

**Broader Impact Concerns:**

There are no concerns.

**Claims And Evidence:**

Yes

**Requested Changes:**

There are no submission-critical adjustments requested. I do have some very minor questions/suggestions on the submission, though. The authors may accept or reject the suggestions are they see fit.
Suggestions:
 - Page 4 paragraph 3
  Replace "This radically removes the need for normalization..."
  with "This removes the need for normalization..."
 - Algorithm 2, Line 7
  Format n_iter as $n_{\mathrm{iter}}$.
 - Page 7 paragraph 3
  Replace:
  "This stochastic optimization approach is highly sought, especially in our..."
  with:
  "This stochastic optimization approach is highly sought after, especially in our..."
 - Page 9 Figure 3 caption
  Replace:
  "...with the outer shell (purple) scatters around."
  with:
  "...with the outer shell (purple) scattered around."
 - Page 11, Last sentence before Eqn. (11)
  Replace:
  "Therefore, any pair (x,y) in a convex $U_u$,..."
  with:
  "Therefore, any pair (x,y) in a geodesically convex $U_u$,..."
 - Page 12, Final paragraph of that page
  Reword
  "...where an island is made up of points that are connected with a finite distance from others."
  The use of the word "with" is confusing.
 - Page 13, First sentence
  Replace:
  "where $q = [-]$ is an equivalence relation with $x \sim x'$..."
  with:
  "where $q = [-]$ is an equivalence relation on $Z$ with $x \sim x'$..."
 - Page 1 of Appendix in the proof of Proposition 1
  Replace:
  "\sum^n_{i = 1}\mu_i \log q_{ij_{i}} upto a constant..."
  with:
  "\sum^n_{i = 1}\mu_i \log q_{ij_{i}} up to a constant..."

Question:
 - Algorithm 1 (Line 10) and Algorithm 2 (lines 1 and 6)
  Why does $D_{\mathrm{glob}}$ have subscript glob? $\mathcal L$ and $\mathcal d$ use subscript glo to stand for global. Why does $D$ differ?
 - Algorithm 1 (Line 6)
  What does "Calculate $n$ distinct $\sigma^2$ by row-wise mean..." mean? Is $\sigma$ the mean or variance of the rows of $D_K$ for each of its rows?
 - Page 10 first paragraph in Section 4.
  In the sentence "Then, in Section 4.2, we proceed our heuristic to construct..." what does "we proceed our heuristic" mean?
 - Page 10 first paragraph of Section 4.1
  Is there some compactness or boundedness assumption needed to ensure that $M$ has finite volume? Otherwise, the uniform distribution may not be defined, as is needed in A2.
 - Page 10 Assumption A1
  Could you please define or else give a reference to the scale, $\lambda_u$?
 - Page 11 Theorem 1 part (b)
  The assumption that $\lambda_u > 0$ is given explicitly, but it seems that it is already required for Assumption A1.
 - Page 12:
  In the statement of Theorem 2, what does the notation $\ceil{*}$ mean?
 - Page 12, Final sentence of the first paragraph in section 4.2
  About the sentence "The advantage of this global construction is its ability to connect two locally disconnected points with a finite global distance". What is meant by the phrase "locally disconnected points?" Is not connected-ness a global property?
 - Page 12, Final paragraph of that page
  The sentence "The space $X_A$ is like a space of islands, as depicted..." What does the phrase "space of islands" mean here? I don't know if the words "space" and "island" are referencing a standard mathematical structure (e.g. metric space) or else the phrase is meant to be understood by its common English meaning.

**Strengths And Weaknesses:**

Strengths:
 - The submission is very well-written and clear.
 - The ability for (i)GLoMAP to capture both local and global structures of data is valuable in applications.
 - The theoretical results are varied, and interesting. Proposition 1 and Theorem 1 give confidence in the training procedure for (i)GLoMAP. The theory and discussion contained in section 4.2 is elegant and gives a nice interpretation for the design of the algorithm.
 - The numerical experiments are varied and extensive.

Weaknesses:
 - Some very minor typos/questions listed below.

---

> ### Author Response · Authors · 2024-10-31
> **Response to Reviewer yBN5 Comments**
>
> **Strength**
>
> Thank you for your positive feedback. We’re glad you found the submission clear and well-written. We appreciate your recognition of our approach to capturing both local and global structures, as well as the theoretical aspects of our work. We’re also pleased that the experiments provided a thorough evaluation. Your comments are very encouraging, and we’re grateful for your insights.
>
> **Requested Changes**
>
> Thank you very much for your thorough review and thoughtful suggestions to improve clarity. We appreciate the time and effort you invested in reviewing the manuscript, and we are grateful for your keen eye for detail. We have incorporated all of your suggestions and highlighted the corresponding parts in deep blue-green for easy reference. We are excited with the enhanced clarity, and we thank you again for your invaluable input!
>
> **Question**
>
> 1. Algorithm 1, Why $D_{\rm glob}$?:
> 	- Thank you for pointing out this notational inconsistency-! Your question made us to realize that  $D_{\rm glob}$ should be fixed by $D_{\rm glo}$ to maintain consistency with $\mathcal{L}_{\rm glo}$ and $d_{\rm glo}$. Now, all $D_{\rm glob}$ 's are changed into the suggested $D_{\rm glo}$ .
> 2. Algorithm 1, $\sigma^2$ by row-wise *mean* ... ?
> 	- We think this is a very good catch. By your question, we realized that it may cause a confusion, and we apologize for it. Your intuition is correct that this quantity $\sigma^2$ should resemble a variance. This $\sigma^2$ in Algorithm 1 line 6 is intended to be the value of $\hat{\sigma}^2_x$ in equation (5). If we think $x$ as like a center of K points, then it looks like the variance (trace of the covariance matrix). Note that (5) is an average of the squared distances.
> 	- To fix the issue, we changed the word from mean to average. Note that this average is over finite *squared* distances, which is equivalent to $\hat{\sigma}^2_x$ in (5).
> 	- We also changed the quantity $\sigma$ in Algorithm 1 (line 6 and line 7) as $\hat{\sigma}$ to make clear the connection between  $\hat{\sigma}^2_x$ in (5) and this quantity.
> 1. Page 10, Section 4, "we proceed our heuristic"?: Thank you so much for pointing out this expression. We have revised it to 'we provide theoretical insights into our heuristic' to improve clarity.
> 2. Page 10, Section 4.1: is compactness or boundedness assumption needed?:
> 	- Thank you for pointing out this important consideration. We agree that, to define a uniform distribution on $M$, it is essential to ensure $M$ has finite volume. We have added a compactness assumption in A2, along with a brief explanation of why this condition was introduced.
> 1. Assumption A1, defining $\lambda_u$?:
> 	- Thank you for pointing out that $\lambda_u$ was not defined. We added the following in A1:", i.e., $g = \lambda_u g^*$, where $g^*$ is a Euclidean metric restricted to a $d$-dimensional subspace."
> 1. Theorem 1 (b) , $\lambda_u>0$ already assumed in A1?:
> 	- Yes, you are right that it is already assumed in A1. Thank you so much for pointing it out, and now we have removed this redundant constraint from Theorem 1 (b).
> 2. Theorem 2, \ceil*?
> 	- Sorry that we did not define this symbol. We meant by it the smallest natural number greater than or equal to. Note that $m$ is between 0 and 1.
> 	- In the manuscript, we have revised Theorem 2 by, $K$ as the smallest natural number greater than or equal to $mn$.
> 3. Locally disconnected points? Is not connected-ness a global property?
> 	- Oh, we can see where the confusion is. Yes, by the shortest path search, more points are connected to each other globally. To avoid confusion, we have deleted this sentence.
> 4. Space and island?
> 	- We acknowledge that our initial wording was somewhat vague. In the revised manuscript, we have clarified the language, avoiding ambiguity in the use of the term "space" and explicitly stating that this is an analogy. We believe the revised version is clearer and reads better:
> 		*Metaphorically speaking, the points in $X_a$ that are within a finite distance from $a$ under $d_a$ can be thought of as composing an "island" centered around $a$ (See $X_a$ in Figure S15, the red-colored part). ... Recalling the island analogy, $X_A$ now contains $n$ islands, as depicted in Figure S15 (in the blue box labeled $X_A$).

---

### Review · Reviewer_JCaC · 2024-10-06

**Summary Of Contributions:**

This work focuses on dimension reduction for visualization, aiming to preserve both local and global distance estimates. The method constructs local distances between sample pairs by considering only the K-nearest neighbors and computes global distances using Dijkstra's algorithm to find the shortest paths between pairs. The final low-dimensional representation is optimized by minimizing the KL divergence between the high- and low-dimensional distances. Stochastic gradient descent is employed instead of using the full dataset, and an inductive approach is introduced using a deep neural network. Theoretical justification is provided for both the local and global distance strategies. Experiments on synthetic and real datasets demonstrate the method's effectiveness compared to existing approaches, with robustness validated by varying K.

**Audience:**

Yes

**Broader Impact Concerns:**

Not applicable.

**Claims And Evidence:**

Yes

**Requested Changes:**

The use of Dijkstra’s algorithm for global distance estimation is interesting, but the presentation makes it hard to fully appreciate the contributions, which seem mostly to combine existing ideas. I suggest major revisions to better highlight the novel aspects, improve the clarity, and include more experiments.

**Strengths And Weaknesses:**

Strengths:

1. The authors address both local and global distance estimates to improve dimension reduction.

2. They provide theoretical justification for the design of the proposed method.

3. Experiments on both synthetic and real datasets demonstrate the method's superiority over existing techniques and robustness across different K values.

Weaknesses:

1. The contribution seems limited. The authors mainly combine existing ideas from Isomap's use of Dijkstra’s algorithm for global distances with UMAP's approach, adding a different rescaling method for local distance estimates.

2. The paper is poorly presented, making it difficult to understand. For example, in Section 1, paragraph 3, the authors are supposed to explain how the algorithm captures local and global details but instead compare the proposed method to UMAP, creating inconsistency. Similar issues appear in paragraph 4. Additionally, the relationship between $\hat{\sigma}_x$ and the global rescaler in Section 4.2 is unclear. Theorem 1 introduces several concepts, but weak sentence connections and scattered logic make it difficult to follow.

3. The authors should include more experiments on realistic datasets, such as FashionMNIST [1], and GoogleNews word vectors [2], etc, to better validate the method’s generalizability.

[1] Xiao et al, Fashion-mnist: a novel image dataset for benchmarking machine learning algorithms. CoRR, abs/1708.07747, 2017.

[2] Mikolov et al, Distributed representations of words and phrases and their compositionality. NeurIPS 2013.

---

> ### Author Response · Authors · 2024-10-31
> **Response to Reviewer JCaC Comments**
>
> **Strength**
> *Thank you for your thoughtful comments and for recognizing our efforts to address both local and global distances in dimension reduction. We appreciate your acknowledgment of the theoretical justification and experimental results that highlighted the method's effectiveness compared to existing techniques.*
>
> **Weakness**
>
> 1. The contribution seems limited.
>
> 	- We understand your concern and apologize that our original submission did not sufficiently highlight the novel aspects of our method. We have rewritten a significant portion of the introduction, including the contribution part. Further details on these changes are provided in the corresponding part addressing the requested revisions below. Thank you for your valuable feedback.
>
> 1.  The paper is poorly presented, making it difficult to understand.
>
> 	- We apologize for the lack of clarity in our writing, and we appreciate your feedback. We revised the paragraphs corresponding to the following comments to improve communication and refined the text's flow to better guide the readers. The changes are in blue in the revised manuscript.
>
> 	1. For example, in Section 1, paragraph 3, the authors are supposed to explain how the algorithm captures local and global details but instead compare the proposed method to UMAP, creating inconsistency.
>
> 	- Sorry that the paragraph is written confusingly. We see that in our original paragraph, our method's explanation was mixed with UMAP comparisons. We have separated explanations for our method and comparison to other methods and made a significant revision in the Introduction.
>
> 	2. Similar issues appear in paragraph 4.
>
> 	- Yes, agree with you. Following your comment, we made a similar edit as above.
>
> 	3. Additionally, the relationship between σ^x and the global rescaler in Section 4.2 is unclear.
>
> 	- We apologize that it was written in a confusing way. We presume that the reviewer is asking about global rescaler in section "4.1" because it is mentioned only in Section 4.1 but not in 4.2. We made a significant edit in the flow of the corresponding area nearing equations 12 and 13, and made it clear that "$\hat{\sigma}_x/C$ is an unbiased estimator of $1/\lambda_x$". We hope the revisions improve readability for the readers.
>
> 	4. Theorem 1 introduces several concepts, but weak sentence connections and scattered logic make it difficult to follow.
> 	- We apologize that it was written confusingly. We admit that there are a lot of pieces of ideas that were mixed and scattered around, losing the readers. We made many changes in section 4.1 surrounding Theorem 1 to improve the flow and cut off many redundant sentences. We appreciate your relevant and honest review comment. All changes are in blue in Section 4.1.
>
> 3. Include more experiments: The authors should include more experiments on realistic datasets, such as FashionMNIST [1], and GoogleNews word vectors [2], etc, to better validate the method’s generalizability.
> 	- Thank you for suggesting applying our method on more real datasets. We have added the result of FMNIST, and discussed the computational limit regarding GoogleNews as follows.
> 	- **FMNIST**: We have applied iGLoMAP on FashionMNIST and the results are in Figure S37, for both the original visualization and the generalization on the test dataset.
> 	- **GoogleNews word vectors**: we admit there is a computational limitation -- with the computational resources we have, applying our method is challenging. As discussed in our original submitted manuscript (page 19), "when the number of points is more than 60,000, we have experienced a significant computational bottleneck." Here the bottleneck happens primarily due to the cost for the shortest path search. Another bottleneck is in the mini-batch sampling scheme because we are sampling according to the probability of $\mu_{j|i}$ as in line 9 in Algorithm 2. We believe that improving this computational bottleneck is an important future direction. From UMAP's perspective, this is the cost of obtaining the global shape and the progression from global to local details.

---

> ### Author Response · Authors · 2024-10-31
> **Response to Reviewer JCaC Comments (2)**
>
> **Requested Changes**
> 1. Better highlight the novel aspects: *Thank you for pointing out an important issue in our presentation. We carefully reconstructed the introduction to emphasize the novel aspects. We summarize our contribution very briefly as follows:*
> 	- For GLoMAP, we have two algorithmic advances:
>
> 		1) enabling UMAP style algorithm for a non-sparse, global distance matrix by using an unbiased loss estimator
>
> 		2) the tempering through $\tau$, which brings an effect of global to local progression, which was unseen in the literature.
>
> 	- For iGLoMAP, the particle-based algorithm is also a useful innovation, as we found the optimization is very challenging when simply using a standard deep learning technique with a neural network mapper for our GLoMAP loss.
> 	- The global distance made with a new local distance estimate itself is also a useful addition, which has better scalability and is different from existing ones.
> 	- As acknowledged by Reviewer X2sp and Reviewer yBN5, we believe that balancing global and local structures, along with the progression from global to local and the method's generalizability, are important contributions. We hope our revisions more effectively communicate these contributions. Thank you for your thoughtful review of our work.
>
> 2. Improve the clarity:
>
> 	- Following the comments in the above "weakness", we have made efforts to maintain unity within each paragraph in the introduction. These changes are in blue and we hope that you find this revised introduction is enjoyable to read and has clear presentation of contributions and ideas. Thank you so much.

---

### Review · Reviewer_X2sp · 2024-10-17

**Summary Of Contributions:**

This paper introduces two novel algorithms for dimensional reduction and visualization: GLoMAP (Global and Local Manifold Approximation and Projection) and iGLoMAP (its inductive version). GLoMAP improves upon existing manifold learning techniques by capturing both global and local structures of high-dimensional data and exhibiting a progression from global to local structure during optimization. iGLoMAP extends GLoMAP's transductive approach to inductive learning by utilizing a deep neural network for generalization to unseen data. The authors demonstrate the effectiveness of these methods through experiments on both simulated and real-world datasets, comparing them to state-of-the-art techniques such as t-SNE and UMAP.

**Audience:**

Yes

**Broader Impact Concerns:**

The paper’s focus on dimensional reduction and visualization methods has broad implications for data analysis, particularly in fields where interpretability and exploratory data analysis are key (biologie, neuroscience, …).

**Claims And Evidence:**

Yes

**Requested Changes:**

Address the two weaknesses:
- a plot of computational time of GLoMAP with t-SNE and UMAP for different datasets (and thus different numbers of data),
- a table of quantitative results such as [1]: compute Silhouette and Trustworthiness scores (which are available in sklearn) for GLoMAP, t-SNE and UMAP.

[1] Van Assel, H., Vayer, T., Flamary, R. and Courty, N., 2024. Snekhorn: Dimension reduction with symmetric entropic affinities. Advances in Neural Information Processing Systems, 36.

**Strengths And Weaknesses:**

Strengths:
- The paper is well written.
- The paper addresses a key issue in manifold learning: capturing both global and local structures. This balance is often lost in methods like t-SNE and UMAP, which tend to favor local structures.
- The progression from global to local structure during optimization provides intuitive and interpretable results, as shown in the experiments.
- The inductive version, iGLoMAP, offers significant scalability and generalizability to unseen data without retraining.
- Experimental results demonstrate competitive or superior performance on various datasets.

Weaknesses:
- The cost of computation is only discussed. It would be interesting to plot a comparison of computational time of GLoMAP with t-SNE and UMAP for different datasets (and thus different numbers of data).
- Some recent papers such as [1] compute Silhouette and Trustworthiness scores (which are available in sklearn). Adding a table of these scores for GLoMAP, t-SNE and UMAP would add quantitative results to the paper.

[1] Van Assel, H., Vayer, T., Flamary, R. and Courty, N., 2024. Snekhorn: Dimension reduction with symmetric entropic affinities. Advances in Neural Information Processing Systems, 36.

---

> ### Author Response · Authors · 2024-10-31
> **Response to Reviewer X2sp Comments**
>
> **Strength**
>
>
> Thank you for your positive comments and recognition of the strengths of our work. We appreciate that you found the paper well-written and that our approach to balancing global and local structures in manifold learning resonated with you. We’re pleased that you found the progression intuitive and are grateful for your acknowledgment of our contributions.
>
> **Weakness**
> 1. The cost of computation is only discussed. It would be interesting to plot a comparison of the computational time of GLoMAP with t-SNE and UMAP for different datasets (and thus different numbers of data).
>
> - Thank you so much for asking for the actual comparison of computational time. We added the requested computational time plot for varying datasets and increasing numbers of data points in Section S8, Figure 20, and referred to it from Concluding Remarks, where we discussed the computational time. All edits are in the olive color.
> - As shown in the figure, the current implementation of GLoMAP is comparatively the slowest. This is due to its reliance solely on Python without optimizations through alternative languages or interfaces like Numba, Cython, or C++. Additionally, the neighborhood search algorithm used is a standard version rather than a faster approximation. We anticipate that this method will attract interest from skilled developers who can further enhance the implementation.
>
> 2. Some recent papers such as [1] compute Silhouette and Trustworthiness scores (which are available in sklearn). Adding a table of these scores for GLoMAP, t-SNE and UMAP would add quantitative results to the paper.
>
> - Thank you so much for pointing us two useful measures and a related paper.
> - As we recognize the relevance of the paper mentioned, we added a reference to [1] in the related work section (Section 2.2 after t-SNE). The added text is in olive color.
> - We have added a new section in the Appendix (Section S7 Additional Measures), to show the results for Silhouette and Trustworthiness scores of GLoMAP and other baseline methods. Because the major baselines are transductive, we focused on comparing GLoMAP (our transductive version). We compared all simulation datasets with increasing $n$. As there are various datasets, a number of data, and the configuration of measure (number of neighbors in the trustworthiness score), we present the results in multiple plots for readability in Figure S26-27. For the case $n=6000$ and $K=1$ for the trustworthiness measure, we present the results in Table S1. All related visualization results of all baselines and GLoMAP are presented in Figure S21 - S25. The added text is in olive color. For the Hierarchical dataset, GLoMAP demonstrates more stable performance than the baselines across all three levels of the hierarchy. For the Spheres dataset, although GLoMAP’s visualization (Figure 27) appears more appealing compared to other baselines, its performance under the two quantitative measures is not as strong. We appreciate that these suggested measures provide valuable insights into aspects of our method not readily perceptible to the human eye.
>
> **Requested Changes**
> 1. a plot of computational time of GLoMAP with t-SNE and UMAP for different datasets (and thus different numbers of data)
>
> - Thank you so much for the suggestion. We have added, the comparison as described above. Thank you so much for suggesting an important comparison.
>
> 2. a table of quantitative results such as (1): compute Silhouette and Trustworthiness scores (which are available in sklearn) for GLoMAP, t-SNE and UMAP.
> - Thank you so much for the suggestion. We have added the comparison of Silhouette and Trustworthiness scores as discussed above.

---

### Decision · Action_Editor_rGaa · 2024-12-03

**Recommendation:** Accept as is

**Comment:**

The reviewers found the idea to combine global and local distance in controlling the low-dimensional projections interesting, and the authors have done a good job in addressing the questions from the reviewers. The manuscript is in a much better shape after revision. All reviewers suggested acceptance.

**Audience:**

Yes, this work is useful for anyone interested in visualizing the representations.

**Claims And Evidence:**

Yes, claims are supported by illustrations, experiments and theoretical results.